# Zoning Productivity Calculation Method of Fractured Horizontal Wells in High-Water-Cut Tight Sandstone Gas Reservoirs under Complex Seepage Conditions

**Benchi Wei** [1,2], **Xiangrong Nie** [1,2,*], **Zonghui Zhang** [3], **Jingchen Ding** [4], **Reyizha Shayireatehan** [5], **Pengzhan Ning** [1,2], **Ding-tian Deng** [1,2] **and Jiao Xiong** [1,2]

1   College of Petroleum Engineering, Xi'an Shiyou University, Xi'an 710065, China;
    21211010041@stumail.xsyu.edu.cn (B.W.); ningpengzhan@163.com (P.N.); 17868486640@163.com (D.-t.D.);
    18706779281@163.com (J.X.)
2   Shaanxi Key Laboratory of Well Stability and Fluid & Rock Mechanics in Oil and Gas Reservoirs, Xi'an Shiyou
    University, Xi'an 710065, China
3   No.2 Gas Production Plant, Sinopec North China Company, Xianyang 712000, China; zhang.zh1208@163.com
4   Exploration and Development Research Institute, Sinopec North China Company, Zhengzhou 450006, China;
    dingjingchen@163.com
5   College of Geology and Mining Engineering, Xinjiang University, Urumqi 830054, China;
    15319012916@163.com
*   Correspondence: nxrcup@163.com; Tel.: +86-18691955927

**Abstract:** Tight sandstone gas reservoirs generally contain water. Studying the impact of water content on the permeability mechanism of tight gas reservoirs is of positive significance for the rational development of gas reservoirs. Selected cores from a tight sandstone gas reservoir in the Ordos Basin were used to establish the variation in its seepage mechanism under different water saturations. The experimental results show that the gas slip factor in tight water-bearing gas reservoirs decreases as the water saturation increases. The stress sensitivity coefficient and the threshold pressure gradient (TPG) increase with increasing water saturation, characterizing the relationships between stress sensitivity coefficients, TPG, permeability, and water saturation. As the water saturation gradually increases, the relative gas phase permeability of tight sandstone gas reservoirs will sharply decrease. When the water saturation exceeds 80%, the gas phase permeability becomes almost zero, resulting in gas almost ceasing to flow. Through the analysis of experimental results, we defined high-water-cut tight sandstone gas reservoirs and analyzed the permeability characteristics of high-water-cut tight sandstone gas reservoirs in different regions. Combining stress sensitivity coefficients and the TPG with permeability and water saturation relationships, we established a zoning productivity calculation method of fractured horizontal wells in high-water-cut tight sandstone gas reservoirs under complex seepage conditions and validated the practicality of the model through example calculations.

**Keywords:** productivity calculation; zoning productivity; high-water-cut tight gas reservoir; fractured horizontal well

## 1. Introduction

Countries around the world possess numerous tight sandstone natural gas reservoirs, displaying significant development potential. For a considerable time to come, they will continue to play a leading role in the field of natural gas development. Non-tight sandstone gas, as an unconventional natural gas resource, is found within tight sandstone reservoirs, typically with a permeability of less than 0.1 mD [1–4]. Such reservoirs generally lack natural productivity or have extremely low productivity, necessitating hydraulic fracturing [5]. With the development and transformation of tight sandstone reservoirs in progress, gas wells often experience water production phenomena [6]. These phenomena affect the seepage

mechanisms, yet there are currently no mature solutions available for quantitative assessment. Therefore, it is imperative to establish a new model for the dual-phase hydraulic fracturing productivity of tight gas reservoirs, considering multiple factors simultaneously, based on a deeper understanding of the impact of water saturation on seepage mechanisms.

When gas flows through porous media, it is influenced by the slip flow effect [7,8]. This is because the interaction forces between gas molecules and solid molecules are relatively weak. At the pipe wall, gas molecules maintain certain movement states and undergo directional motion along the wall through momentum exchange, leading to the directional flow of adjacent gas molecules [9]. The water saturation in tight sandstone gas reservoirs is higher than that in conventional gas reservoirs [10]. Therefore, it is essential to consider the influence of water saturation when studying the slip flow effect in tight gas reservoirs. Some scholars believe that as water saturation increases, the slip flow effect in tight sandstone reduces [11,12]. However, other scholars have different conclusions, suggesting that the slip flow effect increases with water saturation [13,14]. Unfortunately, due to limited research on this topic, the reasons behind this contradiction remain unknown. Hence, it is necessary to conduct a study on the slip flow effect characteristics in tight water-bearing gas reservoirs.

The study of stress sensitivity in reservoirs has a long history. Geertsma [15] defined the rock compressibility coefficient in 1957 to quantitatively describe the phenomenon of pore volume changes caused by changes in reservoir pore pressure. Stress sensitivity has a significant impact on the development of low-permeability oil and gas fields. Scholars have conducted extensive research on the stress sensitivity characteristics of low-permeability tight reservoirs and their impact on gas well productivity [16–20]. As the water saturation of the reservoir gradually increases, the stress sensitivity of tight reservoirs becomes more significant [21–24]. If there is bound water in the rock pores, it will have a certain impact on the strength of argillaceous reservoirs. Bound water forms a water film on the surface of rock particles, reducing pore volume and throat radius and leading to lower permeability. When the stress applied to the rock changes, causing deformation, the permeability of the rock changes more dramatically. Therefore, reservoirs saturated with water exhibit higher stress sensitivity.

The threshold pressure gradient (TPG) is the minimum pressure gradient required to establish a continuous flow of the non-wetting phase in the pores of a rock. The concept of the TPG was first proposed in 1951, and several foreign scholars have since proven through experimental studies the existence of the TPG during fluid seepage processes in low-permeability tight reservoirs [25,26]. The TPG significantly influences gas well productivity. In previous studies, the TPG was often treated as a constant value. However, research has shown that the TPG is not fixed [27,28]. Zafar (2020) conducted a sensitivity analysis of the two main influencing factors of the TPG, namely, permeability and water saturation. The results showed that the TPG is a power function of these two factors. It decreases with increasing permeability but increases with increasing water saturation [29]. Through experimental research, Song (2015) concluded that the TPG in low-permeability tight reservoirs gradually decreases as permeability decreases [30]. Zhu (2022), using experiments with tight sandstone core samples, found that higher water saturation gradually increases the TPG [31]. Wang (2022)'s experimental results indicated that under water conditions, seepage in tight gas reservoirs exhibits nonlinear characteristics and forms the TPG. The TPG for the gas phase has a close power-law relationship with water saturation [32]. For the water-producing gas wells in water-bearing-inclined gas reservoirs considering stress sensitivity, Fu (2022) has developed a new gas well production equation that accurately determines the relationship between gas well production and stress sensitivity and water production [33].

In summary, most scholars have primarily focused on the impact of single-phase gas or single-factor gas–water interactions on well productivity. However, during the production process of gas wells, the fracturing process leads to different flow patterns in various regions, which conventional productivity models have not taken into account. On the other hand, tight gas reservoirs generally contain water, but most researchers have not

quantitatively characterized the impact of water saturation on the permeability mechanisms of gas wells. In this study, the influence of water saturation on permeability mechanisms was analyzed through experiments. Formulas describing the relationship between the stress sensitivity coefficient, initiation pressure gradient, permeability, and water saturation were derived. Through the analysis of experimental results, a definition for tight gas reservoirs with high water saturation was proposed. A multifactor complex flow condition model considering the influence of water saturation on permeability mechanisms was established for fractured horizontal wells in tight gas reservoirs. This model was then compared with the unobstructed flow rates obtained from productivity tests to validate its accuracy. The new model presented in this paper provides a basis for predicting the productivity of tight sandstone gas reservoirs.

## 2. Experimental Study on the Seepage Mechanism of Tight Cores

Dongsheng gas field is a tight sandstone gas reservoir located in the northeastern Ordos Basin, northern China. The main gas-bearing horizon is the Shihezi Formation and the Shanxi Formation with depths from 3000 to 3600 m. The predominant lithology of the reservoir rocks is lithic quartz sandstone and quartz sandstone, with a porosity ranging from 5.0% to 18.8% and permeability ranging from 0.20 to 3.99 mD. The liquid–gas ratio during production ranges from 5.7 to 7.4 $m^3/10^4$ $m^3$. This gas field is characterized as a lithologic, lithologic–structural, and structural–fracture composite gas reservoir.

### 2.1. Characterization of Gas Slip Effects in Tight Water-Bearing Gas Reservoirs

The gas slip effect refers to the gas in the low porosity porous media flow. When the average free range of gas molecules (gas molecules and other molecules between two successive collisions, through the average value of straight-line distance) is close to the radius of the pore, gas molecule and media pore wall collision increases and the wall of each molecule is in a state of motion. Therefore, in the wall of the tube, a slip flow is generated on the tube wall, resulting in a non-Darcy flow (non-linear flow) when measuring permeability with gas, making the measured gas permeability greater than the actual permeability of the phenomenon, as was discovered by Klinkenberg in 1941 in a test [34].

The corrected model for gas permeability measurements:

$$K = K_\infty \left(1 + \frac{b}{p_m}\right) \tag{1}$$

where $K$ is the gasometric permeability, $K_\infty$ is the absolute permeability, $b$ is the slip factor, and $p_m$ is the average pore pressure.

Natural cores from the Dongsheng gas field were used in the experiments to study the gas slip effects of tight cores. Relevant basic petrophysical parameters of the samples are listed in Table 1. To conduct the experiments, the core samples were initially dried for 24 h and their dry weights were recorded. Subsequently, we saturated the samples using a combination of vacuum suction and pressure injection methods to calculate the pore volume and porosity. Finally, the samples were placed in a core holder, and high-purity nitrogen gas was used for displacement to achieve the desired experimental water saturation.

**Table 1.** Petrophysical parameters of the core.

| Serial Number | Well Number | Core Number | Sample Depth (m) | Length (cm) | Diameter (cm) | Permeability (mD) |
|:---:|:---:|:---:|:---:|:---:|:---:|:---:|
| 1 | X1 | 5-1/57 | 2819.80 | 6.058 | 2.522 | 0.1596 |
| 2 | X2 | 2-43/49 | 2994.97 | 5.693 | 2.520 | 1.4145 |
| 3 | X3 | 2-7/33 | 2872.18 | 6.208 | 2.510 | 0.2348 |
| 4 | X4 | 1-3/47 | 2941.38 | 5.026 | 2.520 | 1.0835 |

The prepared rock samples with varying water saturations were individually placed into an inert gas permeability measurement system for low-permeability rocks. This allowed us to measure the permeability of the samples under different confinement pressures and injection pressures. To ensure the reliability of the data, this article conducted tests on four sets of samples for each water saturation level and obtained experimental results by measuring their slip factor. For specific data, please refer to Table 2.

**Table 2.** Slip factor for different perimeter pressures; different water saturations for different core conditions.

| Well Number | Water Saturation/% | Pressurization | | | |
|---|---|---|---|---|---|
| | | 3 | 5 | 10 | 15 |
| X1 | 0 | 0.17082 | 0.17706 | 0.17794 | 0.17984 |
| | 15 | 0.14246 | 0.14291 | 0.10449 | 0.10040 |
| | 30 | 0.08072 | 0.11535 | 0.05944 | 0.04743 |
| | 40 | 0.00313 | 0.00119 | 0.00006 | 0.00007 |
| X2 | 0 | 0.08123 | 0.08420 | 0.08462 | 0.08552 |
| | 15 | 0.06774 | 0.01291 | 0.04969 | 0.04774 |
| | 30 | 0.03839 | 0.00198 | 0.02826 | 0.02255 |
| | 40 | 0.00149 | 0.00057 | 0.00003 | 0.00003 |
| X3 | 0 | 0.14961 | 0.15508 | 0.15585 | 0.15751 |
| | 15 | 0.12477 | 0.02378 | 0.09151 | 0.08793 |
| | 30 | 0.07070 | 0.00365 | 0.05206 | 0.04154 |
| | 40 | 0.00274 | 0.00104 | 0.00006 | 0.00006 |
| X4 | 0 | 0.08894 | 0.09219 | 0.09265 | 0.09363 |
| | 15 | 0.07417 | 0.01414 | 0.05440 | 0.05227 |
| | 30 | 0.04203 | 0.00217 | 0.03095 | 0.02469 |
| | 40 | 0.00163 | 0.00062 | 0.00003 | 0.00004 |

The data from Figure 1 (X1–X4) clearly demonstrate that as the water saturation of the rock samples increases, the slip factor gradually decreases. This indicates that with the increasing water saturation of the rock samples, the gas slip effect diminishes progressively, and at a water saturation of 40%, the gas slip effect is almost completely inhibited.

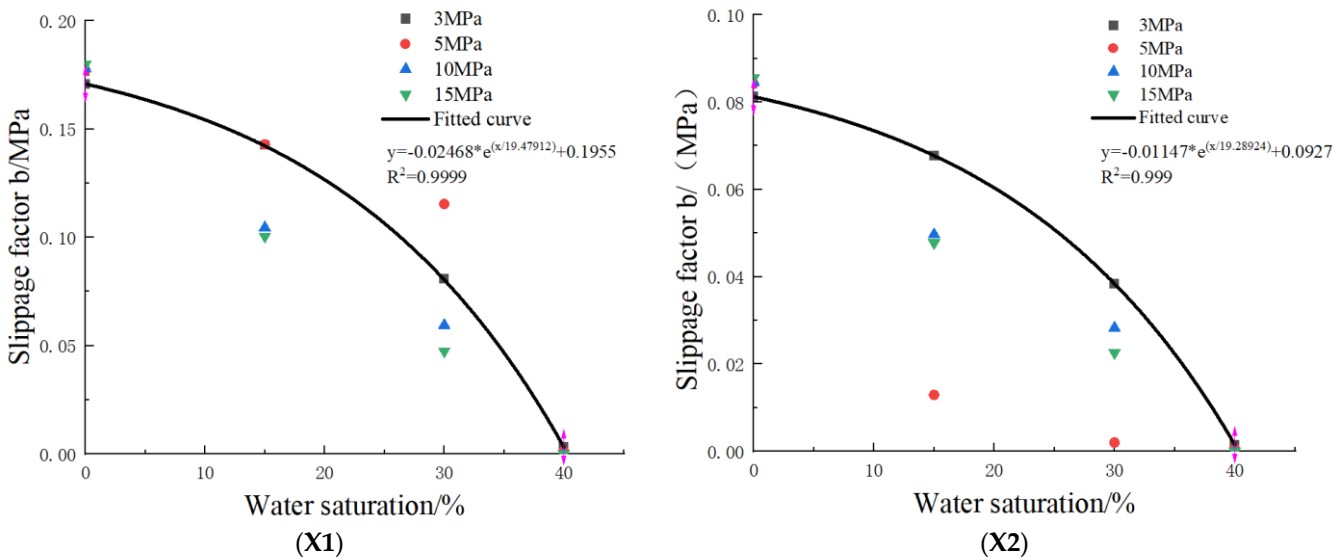

**Figure 1.** *Cont.*

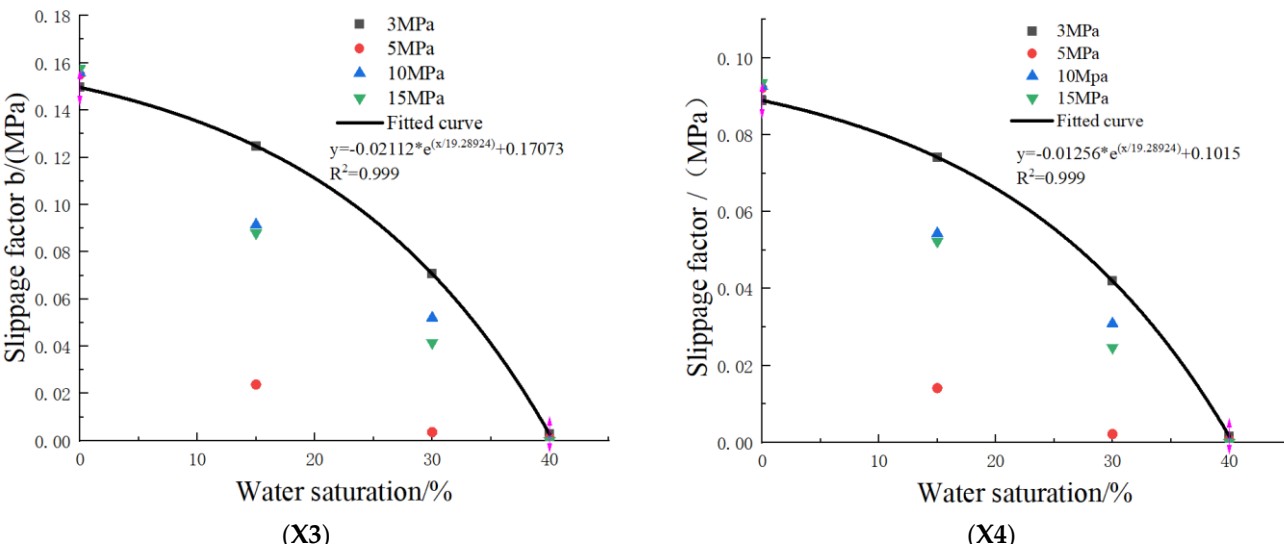

**Figure 1.** The curves depicting the variation in slip factors in wells (**X1**–**X4**) with changing water saturation.

*2.2. Stress-Sensitive Characterization of Tight Water-Bearing Gas Reservoirs*

Rock stress sensitivity refers to the extent to which the permeability of rock changes under different effective stress conditions. There are noticeable differences in stress sensitivity among different types of reservoir rocks. Stress sensitivity is assessed by applying external stress from overlying strata and internal fluid pressure. Therefore, measuring the permeability of rocks under different confining pressure conditions to evaluate their stress sensitivity is one of the most commonly used methods in laboratory experiments. This method complies with the petroleum industry standard SY/T5358-2010, titled "Evaluation Method for Reservoir Sensitivity Flow Experiments". The conventional stress sensitivity evaluation experimental method involves altering the confining pressure of the core under atmospheric pressure conditions and subsequently measuring the permeability under the corresponding pressures.

This study utilized nine core samples from a tight gas reservoir in the Ordos Basin, and the basic rock properties of these samples are listed in Table 3. In the stress sensitivity experiments conducted under confined water conditions, the establishment of water saturation within the cores was crucial. Firstly, the cores were dried and subjected to vacuum treatment to simulate the saturation of formation water, and the mass of the saturated cores was measured. Subsequently, the cores were placed in core holders, and gas displacement was conducted. During this operation, appropriate displacement pressures were selected based on the core's porosity and permeability data. Throughout the displacement process, the cores were taken out and rotated at both ends to ensure an even distribution of confined water. The mass of the cores was repeatedly measured until the desired level of confined water saturation was achieved. Subsequently, stress sensitivity experiments were conducted under these water saturation conditions.

Conventional stress sensitivity evaluation experiments, involving alterations in confining pressure, were performed according to the petroleum industry standard SY/T5358-2010, titled "Evaluation Method for Reservoir Sensitivity Flow Experiments." Different confining pressure levels (5, 10, 15, 20, 25, 30, 35, 40, 35, 30, 20, 15, 10, and 5 MPa) were set, and changes in core permeability were measured. Table 3 presents the results of stress sensitivity experiments at various water saturation levels.

The stress sensitivity coefficient (Ss) is calculated using the following formula:

$$S_s = \left[1 - (k_i/k^*)^{1/3}\right]\lg(\sigma_i/\sigma^*) \tag{2}$$

where $\sigma_i$ denotes the confining pressure, $k_i$ is the permeability of the core at different confining pressures, $\sigma^*$ is the reference stress point, and $\sigma_i$ is the permeability corresponding to $\sigma^*$.

**Table 3.** The results of stress sensitivity experiments at different water saturation levels.

| Well Number | Core Number | Length (cm) | Diameter (cm) | Initial Permeability (mD) | Permeability at 40 Mpa (mD) | $S_w$ (%) | Permeability Damage at 40 Mpa (%) | Stress Sensitivity Coefficient |
|---|---|---|---|---|---|---|---|---|
| X1 | 5-1/57 | 6.058 | 2.522 | 0.0967 | 0.0047 | 0 | 95.19 | 1.46 |
|  |  |  |  | 0.0605 | 0.0018 | 45 | 97.01 | 1.69 |
|  |  |  |  | 0.0266 | 0.0005 | 55 | 98.25 | 1.94 |
|  |  |  |  | 0.0100 | 0.00002 | 65 | 99.77 | 2.91 |
| X1 | 6-14/18 | 5.264 | 2.512 | 0.1944 | 0.0172 | 0 | 91.18 | 1.17 |
|  |  |  |  | 0.1157 | 0.0082 | 45 | 92.93 | 1.27 |
|  |  |  |  | 0.0807 | 0.0038 | 55 | 95.33 | 1.47 |
|  |  |  |  | 0.0369 | 0.0003 | 65 | 99.22 | 2.33 |
| X2 | 2-32/49 | 5.837 | 2.517 | 0.9215 | 0.2319 | 0 | 72.83 | 0.63 |
|  |  |  |  | 0.2886 | 0.0738 | 45 | 74.44 | 0.66 |
|  |  |  |  | 0.1578 | 0.0235 | 55 | 85.09 | 0.92 |
|  |  |  |  | 0.0810 | 0.0049 | 65 | 93.96 | 1.35 |
| X3 | 2-7/33 | 6.208 | 2.510 | 0.1586 | 0.0208 | 0 | 84.86 | 0.91 |
|  |  |  |  | 0.1622 | 0.0214 | 45 | 86.80 | 0.97 |
|  |  |  |  | 0.1338 | 0.0138 | 55 | 89.68 | 1.09 |
|  |  |  |  | 0.0861 | 0.0044 | 65 | 94.91 | 1.43 |
| X4 | 1-3/47 | 5.130 | 2.523 | 0.8514 | 0.3840 | 0 | 48.90 | 0.32 |
|  |  |  |  | 0.3758 | 0.1849 | 45 | 50.81 | 0.34 |
|  |  |  |  | 0.1533 | 0.0667 | 55 | 56.48 | 0.40 |
|  |  |  |  | 0.0423 | 0.0145 | 65 | 65.64 | 0.51 |
| X5 | 1-5/47 | 5.026 | 2.520 | 0.5767 | 0.2266 | 0 | 53.70 | 0.37 |
|  |  |  |  | 0.3377 | 0.1487 | 45 | 55.97 | 0.39 |
|  |  |  |  | 0.0453 | 0.0085 | 65 | 81.25 | 0.81 |
| X6 | 4-46/55 | 5.569 | 2.519 | 0.7001 | 0.2888 | 0 | 58.74 | 0.43 |
|  |  |  |  | 0.4576 | 0.1247 | 45 | 72.75 | 0.63 |
|  |  |  |  | 0.1144 | 0.0091 | 65 | 92.08 | 1.22 |
| X7 | 1-5/38 | 6.125 | 2.472 | 0.7491 | 0.3430 | 0 | 49.22 | 0.33 |
|  |  |  |  | 0.2535 | 0.1233 | 45 | 51.38 | 0.35 |
|  |  |  |  | 0.0194 | 0.0020 | 65 | 89.77 | 1.10 |
| X8 | 3-45/51 | 6.270 | 2.443 | 0.2292 | 0.0692 | 0 | 69.81 | 0.58 |
|  |  |  |  | 0.3038 | 0.0726 | 45 | 76.08 | 0.69 |
|  |  |  |  | 0.2311 | 0.0599 | 55 | 74.08 | 0.65 |
|  |  |  |  | 0.1492 | 0.0289 | 65 | 80.65 | 0.79 |

The dimensionless net confining pressure and dimensionless permeability can be fitted into the following relationship:

$$\frac{K}{K_i} = \left( \frac{p_c - p_p}{p_c - p_i} \right)^{-s_s} \tag{3}$$

where $K_i$ represents the initial permeability.

Taking core X1 as an example, the experimental data indicate that it exhibits different stress sensitivities under various levels of confined water saturation. For each distinct level of confined water saturation, the stress sensitivity coefficient ($S_s$) of core X1 varies, and these coefficients can be normalized based on stress sensitivity experimental data. According to the results shown in Figure 2, it can be observed that the stress sensitivity coefficients ($S_s$) of the nine cores involved in the experiment change under different levels

of confined water saturation and water saturation conditions. Further analysis reveals that, at zero water saturation, core X1 exhibits the lowest stress sensitivity, but as water saturation increases, its sensitivity gradually intensifies. As permeability stabilizes, an increase in confined water saturation leads to a significant exponential growth trend in the stress sensitivity coefficient of X1.

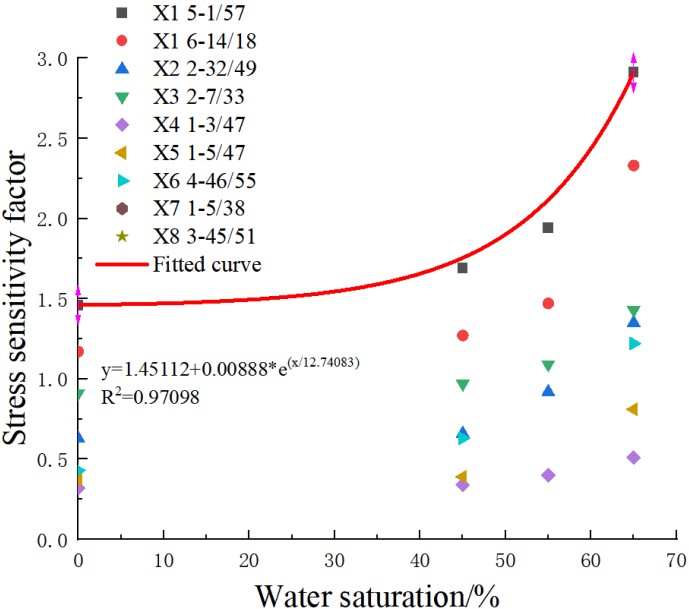

**Figure 2.** Relationship between water saturation and the stress sensitivity coefficient.

In the experimental process, in order to study the influence of permeability on stress sensitivity at a specific bound water saturation, the bound water saturation values established for each core sample were kept relatively consistent. The difference between the maximum and minimum values for each of the nine core samples at each bound water saturation point did not exceed 3%. Figure 3 illustrates the relationship between permeability and the stress sensitivity coefficient for the nine core samples at a specific bound water saturation.

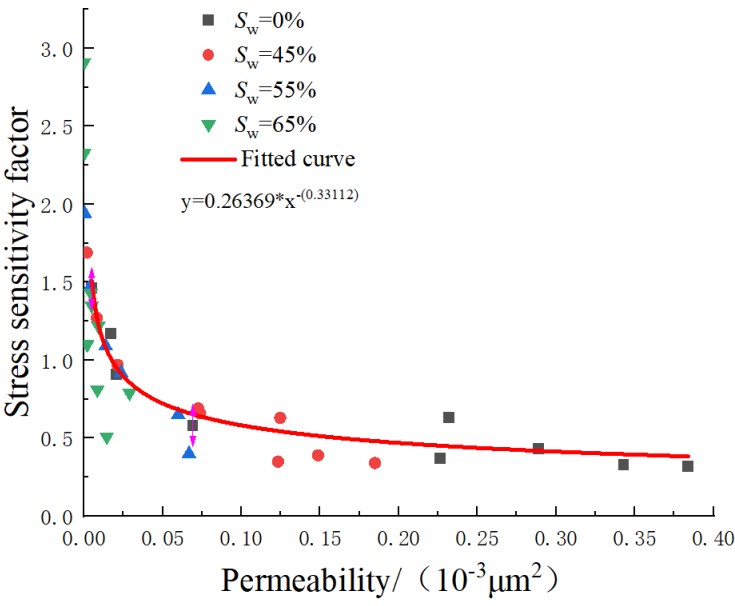

**Figure 3.** The influence of permeability on the stress sensitivity coefficient.

When the water saturation is at 0%, an exponential relationship is observed between the stress sensitivity coefficient and permeability. As permeability decreases, the stress sensitivity coefficient shows an increasing trend. Similarly, at water saturations of 45%, 55%, and 65%, there continues to be an exponential relationship between permeability and the stress sensitivity coefficient. Therefore, it can be reasonably assumed that the relationship between permeability and the stress sensitivity coefficient can be described by an exponential function:

$$S_S = aK^{-b} \tag{4}$$

The relationships between the exponent coefficients *a* and *b* in Equation (2) and water saturation are arranged as shown in Figures 4 and 5.

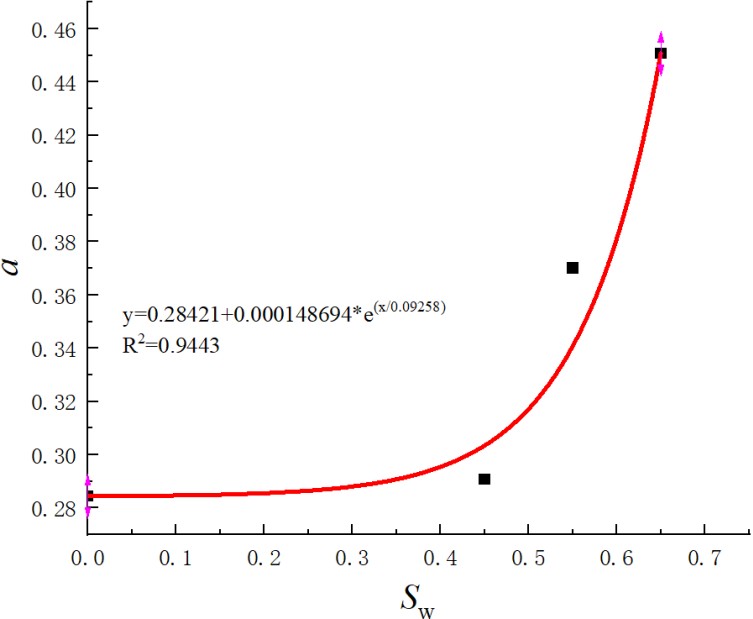

**Figure 4.** Relation between water saturation $S_w$ and a.

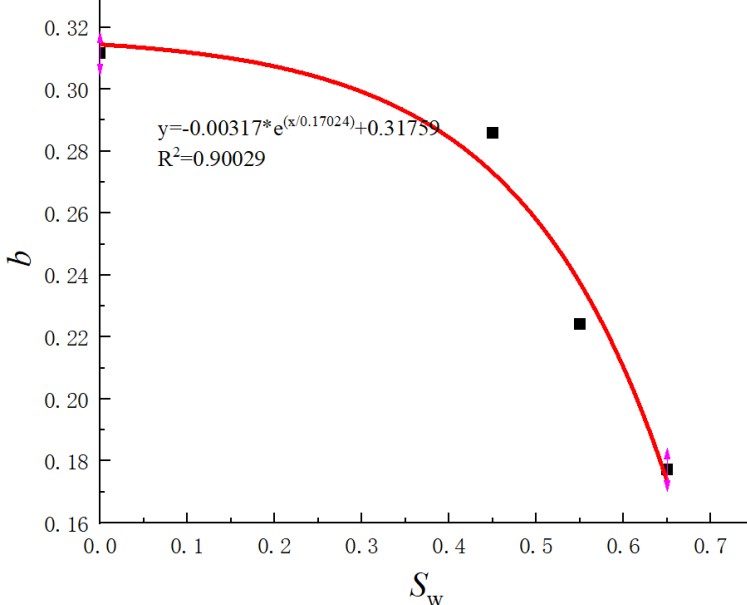

**Figure 5.** Relation between water saturation $S_w$ and b.

The relationship between the multiplicative power coefficient *a* and the bound water saturation $S_w$ can be obtained from Figure 4:

$$a = 0.28421 + 0.000148694 * e^{(S_w/0.09258)} \tag{5}$$

The relationship between the multiplicative power index *b* and the bound water saturation $S_w$ can be obtained from Figure 5:

$$b = -0.0317 * e^{(S_w/0.17024)} + 0.31759 \tag{6}$$

By substituting Equations (3) and (4) into Equation (2), we obtain:

$$S_S = (0.28421 + 0.000148694 * e^{(S_w/0.09258)}) * K^{-(-0.0317*e^{(S_w/0.17024)}+0.31759)} \tag{7}$$

where $S_s$ is the stress sensitivity factor for bound water conditions; $S_w$ is the binding water saturation; and *K* is the absolute permeability, $10^{-3}$ μm². 

Formula (5) represents the stress sensitivity relationship of low-permeability tight gas reservoirs under confined water conditions. Since conducting stress sensitivity experiments under confined water conditions involves complex procedures and significant difficulties, the significance of Formula (5) lies in its ability to determine the reservoir's stress sensitivity under confined water saturation conditions using the permeability and water saturation values of the tight gas reservoir. In this case, this formula provides a convenient method, thereby avoiding the extensive use of manpower, resources, and financial investment required for experiments. Therefore, this approach can better assist researchers in understanding the mechanical properties of reservoir rocks and provide more reliable support for the development of tight gas reservoirs.

### 2.3. Characteristics of the TPG in Tight Water-Bearing Gas Reservoirs

The Jamin effect is one of the main reasons for the generation of the TPG in tight sandstone gas reservoirs. In porous media within tight reservoirs, there are pores of different sizes and narrow throats connecting these pores, forming a complex pore network. When bubbles flow from larger pores into narrower throats, they encounter resistance and need to overcome the deformation of the bubbles to continue flowing. This phenomenon is known as the Jamin effect. When the external driving pressure cannot overcome the capillary pressure, bubbles accumulate and block the throats. Over time, bubbles accumulate energy until gas and water phases can break through these constraints and start flowing. Therefore, a necessary condition for gas flow is that the pressure difference between the two sides of the bubble's surface reaches a certain level. This effect is crucial for understanding the productivity of gas reservoirs.

Five core samples from a tight sandstone gas reservoir in the Ordos Basin were selected, with the physical properties of the core samples as shown in Table 4. Experiments were conducted under different water saturation conditions to determine the TPG.

**Table 4.** Basic rock properties for the TPG testing of the core samples.

| Well Number | Diameter (cm) | Length (cm) | Atmospheric Pressure Porosity (%) | Atmospheric Pressure Permeability ($\times 10^{-3}$ μm²) | Remarks |
|---|---|---|---|---|---|
| X1 | 2.534 | 5.968 | 5.51 | 0.05 | Conventional Core Samples |
| X2 | 2.536 | 6.210 | 6.98 | 0.17 | Conventional Core Samples |
| X3 | 2.538 | 4.254 | 8.46 | 0.53 | Conventional Core Samples |
| X4 | 2.535 | 6.041 | 10.63 | 1.24 | Conventional Core Samples |
| X5 | 2.538 | 6.084 | 10.95 | 3.94 | Fractured Core Samples |

The sentence describes the permeability characteristics under different water saturation conditions, as shown in Figure 6, revealing an important phenomenon: as the water saturation of the rock core samples increases, the TPG gradually increases. Especially

when the permeability remains constant, with the increasing saturation of the bound water, the TPG exhibits a pronounced exponential growth trend. This discovery highlights the significant influence of water saturation on rock permeability and startup characteristics.

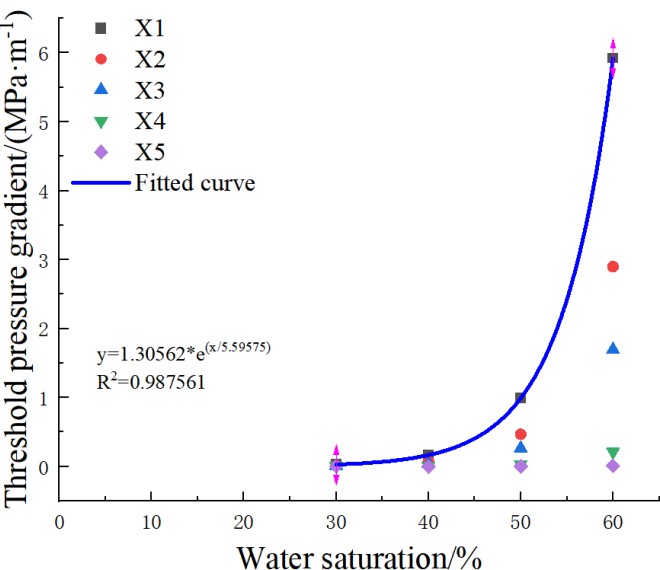

**Figure 6.** TPG versus water saturation curve.

To study the influence of permeability on stress sensitivity under a certain bound water saturation, in the experimental process, it is ensured that the bound water saturation values established for each core sample are essentially consistent. Furthermore, the difference between the maximum and minimum values at each water saturation point does not exceed 3%. As shown in Figure 7, the study investigated the relationship between permeability and initiation pressure gradient for these five core samples at different water saturation levels.

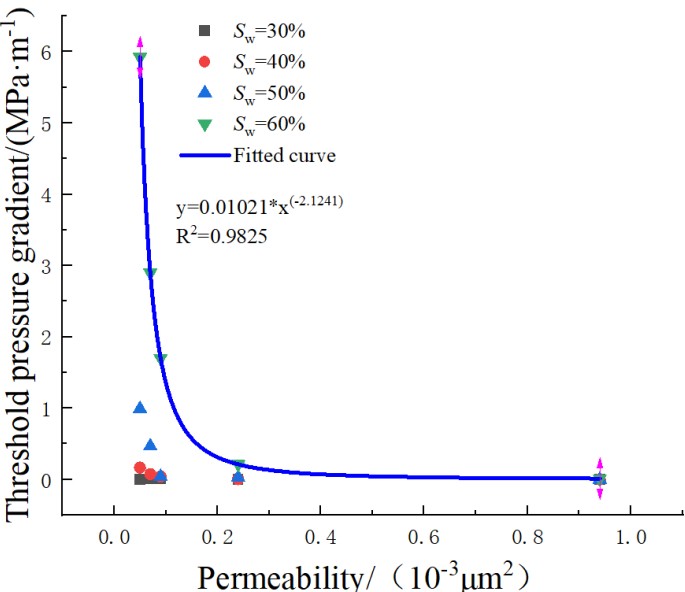

**Figure 7.** The impact of permeability on the TPG.

According to the results of the pressure gradient testing experiment, it is evident that there is a linear relationship between the reciprocal of the core sample's permeability and the startup pressure gradient, while the coefficient "*a*" exhibits an exponential relationship with water saturation ($S_w$) (as detailed in Figure 8). Additionally, the coefficient "*b*" shows

a linear relationship with the reciprocal of water saturation (as detailed in Figure 9). By fitting the test data from the five core samples, a relationship formula for the TPG in tight sandstone gas reservoirs was derived. This formula is related to both permeability and water saturation.

$$\lambda = ak^{(-b)} = (1.41741 * 10^{-6} * e^{(S_w/5.36345)})K^{-(86.25654*S_w^{(-0.91146)})} \tag{8}$$

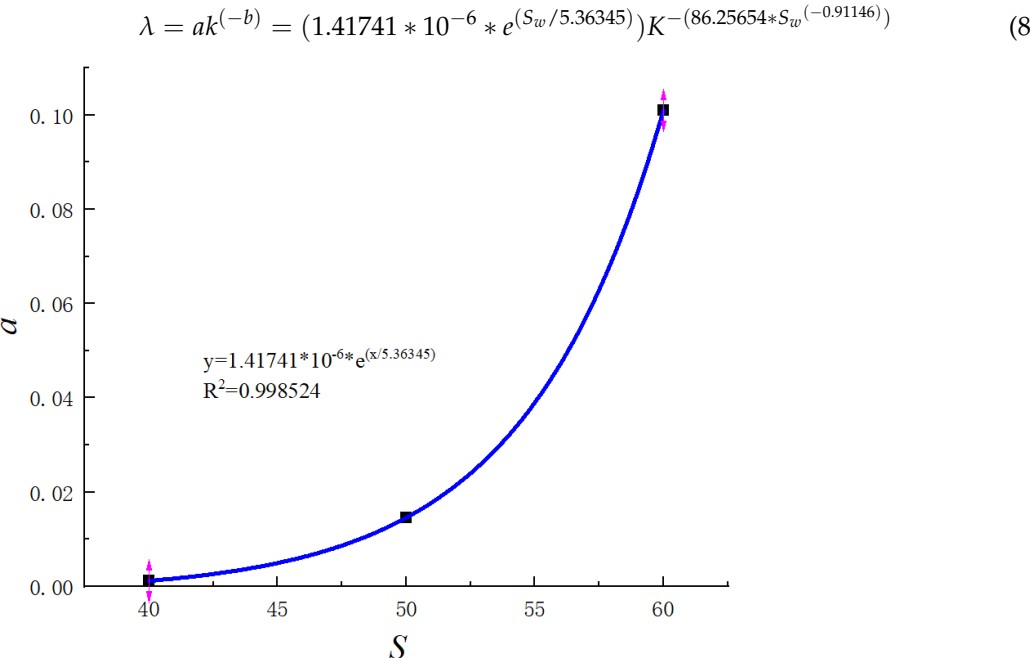

**Figure 8.** Curve of water saturation versus TPG coefficient a.

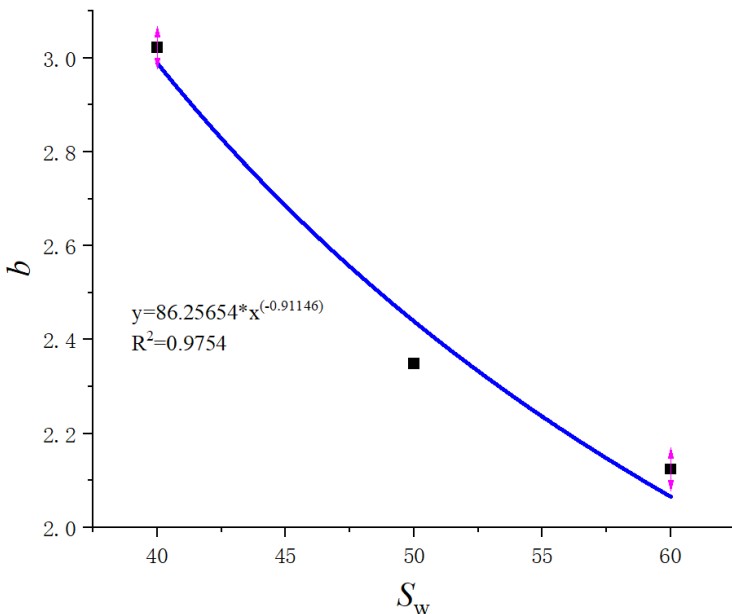

**Figure 9.** Curve of water saturation versus TPG coefficient b.

### 2.4. Two-Phase Seepage Characteristics in Tight Water-Bearing Gas Reservoirs

In order to study the gas–water co-permeability characteristics of low-permeability reservoirs, this article selected seven tight sandstone core samples for gas-driven water experiments. The experimental procedure is illustrated in Figure 10 and includes the following steps:

First, the core samples are placed in core holders, and appropriate displacement pressures are determined based on the physical properties of the core and water permeability

test data. Next, the gas pressure inside the intermediate container is stabilized at the set displacement pressure, and both the inlet and outlet of the core are opened to initiate the gas displacement water experiment. During the experiment, the weight of water exiting the core and the gas flow rate are continuously measured using a precision balance and a gas mass flow meter, respectively. Simultaneously, a computer collects data on cumulative water and gas production at different time intervals. Finally, using pressure, water production, and gas production data at different time points, gas–water relative permeability curves under various water saturation conditions during gas displacement water experiments are calculated using software for processing. These curves reflect the characteristics of gas–water co-seepage and the seepage behavior of gas and water. Table 5 summarizes the experimental results data.

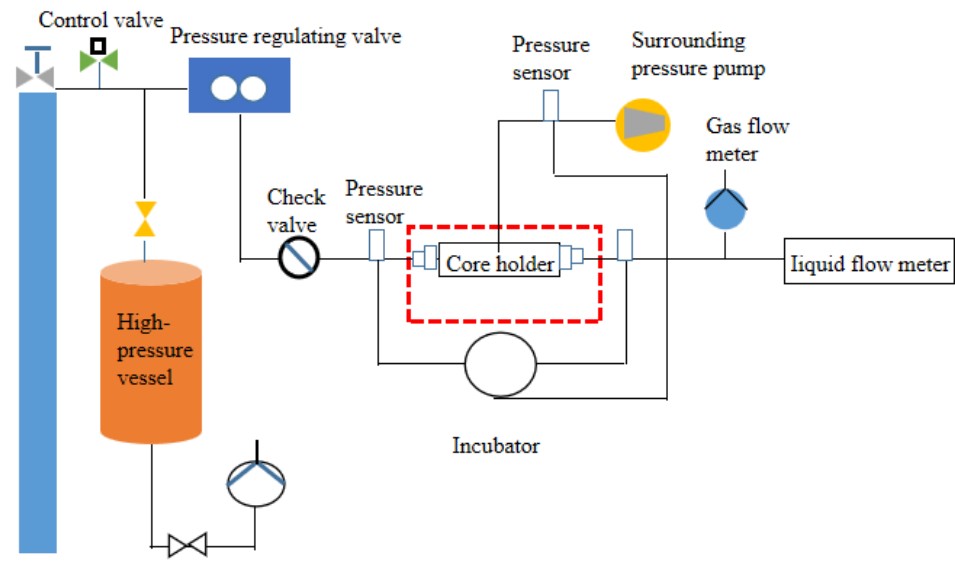

**Figure 10.** Gas–water phase permeation experiment flowchart.

**Table 5.** Gas–water phase permeation experiment results.

| Number | Well Number | Length (cm) | Diameter (cm) | Gas Permeability by Gas Measurement $\times 10^{-3}$ $\mu m^2$ | Porosity (%) | Test Pressure Differential (Mpa) | Formation Water Permeability by Water Measurement $\times 10^{-3}$ $\mu m^2$ | Water Saturation (%) | Gas Relative Permeability |
|---|---|---|---|---|---|---|---|---|---|
| 1 | | | | | | 3.74 | | 64.21 | 0.3716 |
| 2 | X1 | 6.7 | 2.403 | 0.392 | 10.8 | 4.45 | 0.007 | 63.36 | 0.4698 |
| 3 | | | | | | 5.42 | | 59.77 | 0.5934 |
| 4 | | | | | | 1.54 | | 58.96 | 0.5180 |
| 5 | X2 | 5.537 | 2.475 | 1.997 | 10.8 | 1.84 | 0.044 | 54.35 | 0.7120 |
| 6 | | | | | | 2.17 | | 49.91 | 0.8898 |
| 7 | | | | | | 7.11 | | 61.93 | 0.4750 |
| 8 | X3 | 5.731 | 2.471 | 0.199 | 9.9 | 8.78 | 0.002 | 57.92 | 0.6111 |
| 9 | | | | | | 10.64 | | 51.92 | 0.6625 |
| 10 | | | | | | 1.08 | | 63.95 | 0.6592 |
| 11 | X4 | 6.27 | 2.443 | 1.028 | 16.6 | 1.31 | 0.087 | 61.99 | 0.7192 |
| 12 | | | | | | 1.52 | | 61.24 | 0.7769 |

Figure 11 illustrates the gas-water phase permeability curves for each rock sample at different pressure gradients. The experimental results indicate:

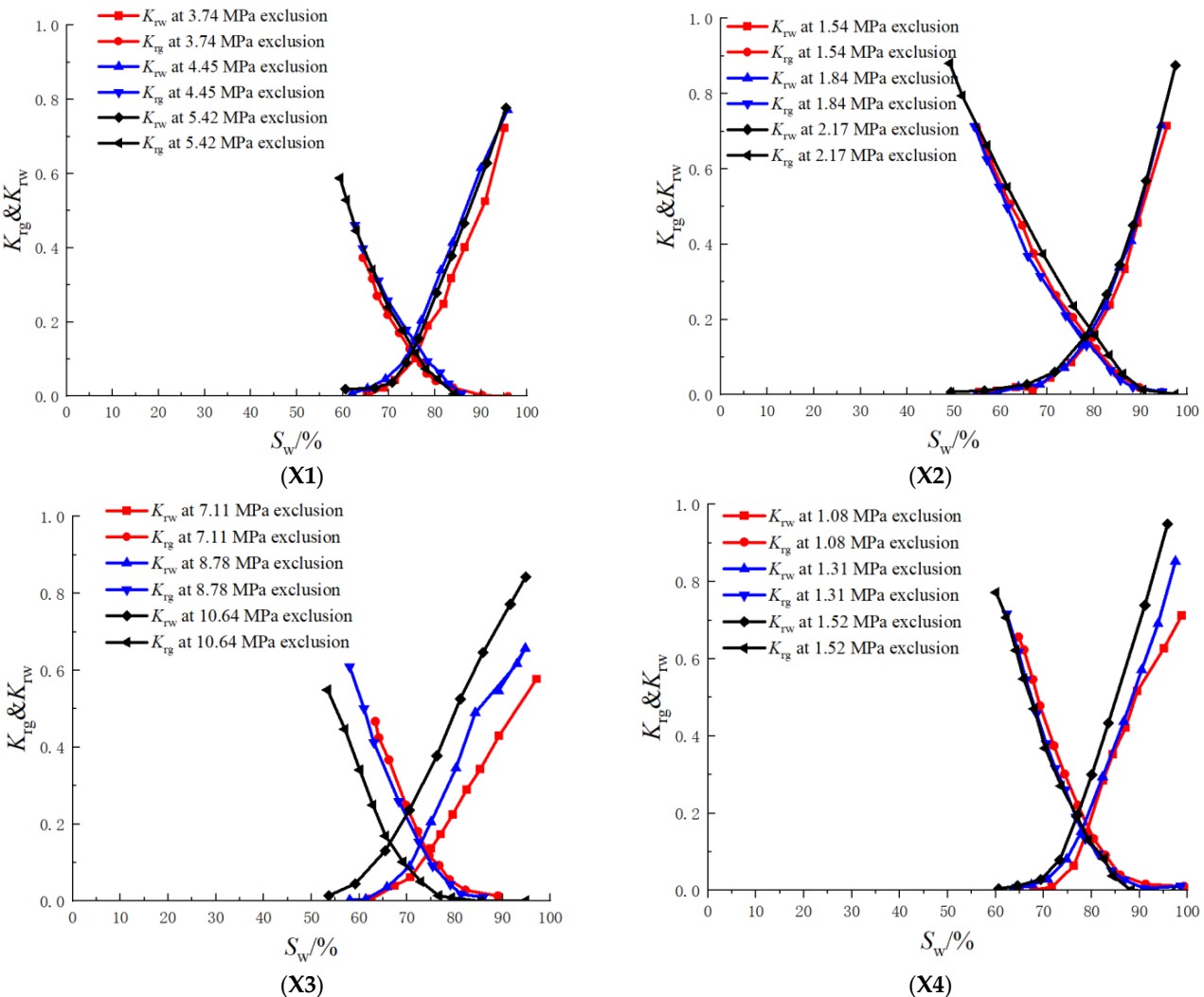

**Figure 11.** Gas–water phase permeation curves for (**X1–X4**) rock sample at various pressure gradients.

Pressure gradients have a significant impact on the permeation ability of gas and water phases. With an increase in pressure gradient, the relative permeability of the gas phase exhibits a leftward trend. This implies that under the same water saturation conditions, higher displacement pressure gradients lead to a decrease in the relative permeability of the gas phase. This effect is more pronounced at lower water saturations and gradually diminishes as water saturation increases to around 75% or so. On the other hand, the relative permeability of the water phase increases with an increase in pressure gradient, especially under conditions with higher water saturation, where the impact is more significant.

Regardless of the pressure gradient conditions, as water saturation increases, the relative permeability of the gas phase in tight sandstone gas reservoirs decreases sharply. When the water saturation reaches 80% or higher, the gas phase permeability becomes almost negligible, resulting in minimal gas penetration.

These findings emphasize the significant influence of pressure gradients and water saturation on the gas–water co-permeation characteristics of tight sandstone gas reservoirs, especially in low-permeability reservoirs. This has important practical implications for the development and management of gas reservoirs.

## 3. Derivation of Productivity Equation for Fractured Horizontal Wells in High-Water-Cut Tight Gas Reservoirs

In high-water-cut tight gas reservoirs, the gas flow patterns vary in different regions due to hydraulic fracturing, accompanied by different flow effects. This section first defines

a high-water-cut tight gas reservoir through the analysis of experimental results and then derives the productivity equations for different regions. These research findings contribute to a better understanding and management of the development process in high-water-cut tight gas reservoirs, aiming to enhance reservoir productivity and efficiency.

### 3.1. Definition of High-Water-Cut Tight Gas Reservoir

(1)  Gas slippage effect

The analysis of Figure 12a leads to the following conclusion: as the water saturation increases, the slip factor gradually decreases. This is because as the average free path of the gas molecules approaches the capillary diameter, the collision frequency between the gas molecules and the pore walls of the porous media increases, leading to the occurrence of sidewall slip flow phenomena. This in turn leads to a continuous increase in gas velocity. Therefore, when gas is used as a permeability measurement medium, non-Darcy flow phenomena will inevitably occur, causing the measured permeability to be significantly higher than the true permeability.

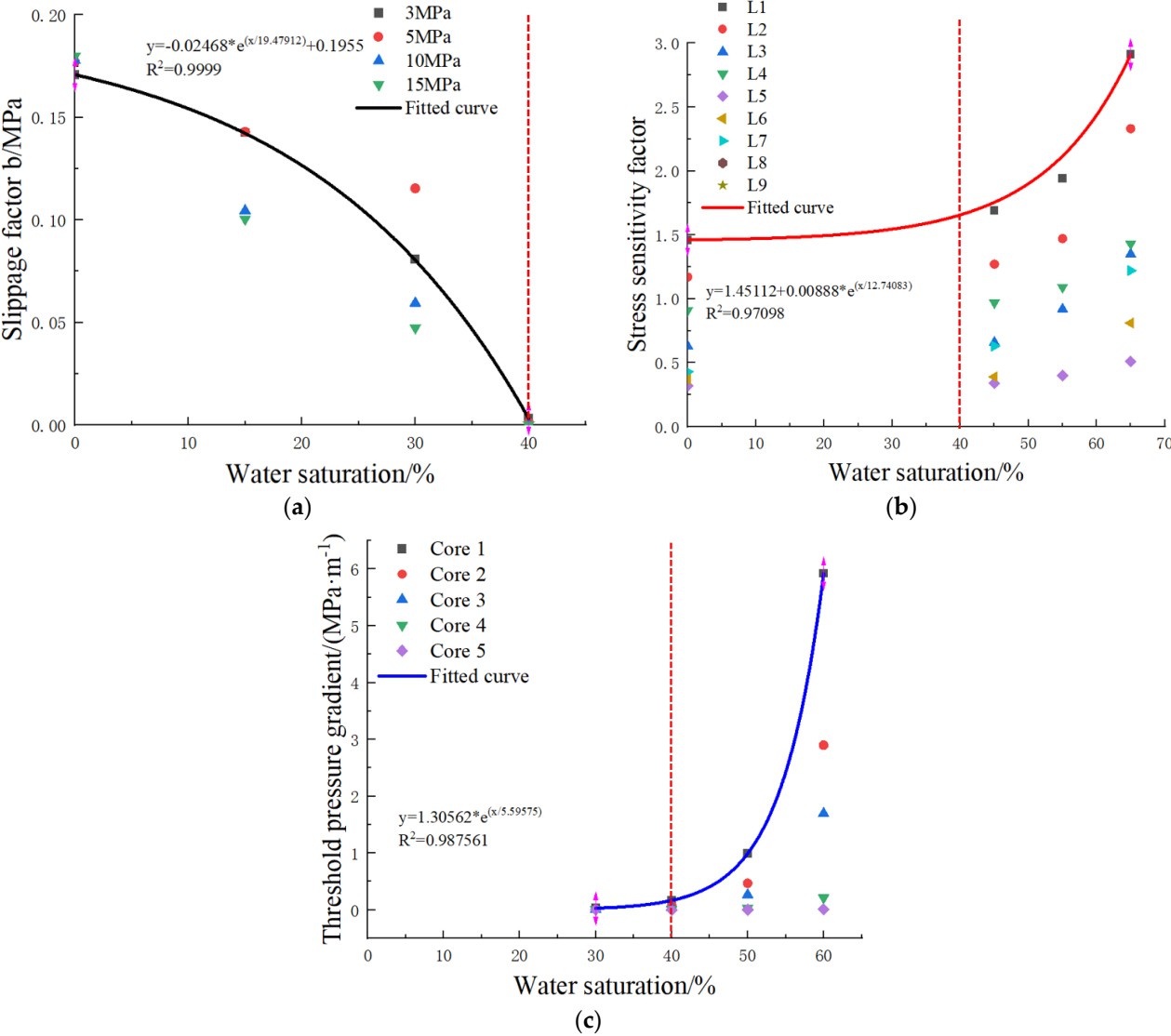

**Figure 12.** The permeability law of tight water-bearing gas reservoirs. (**a**) The permeability law of tight water-bearing gas reservoirs; (**b**) the stress sensitivity characteristics of tight water-bearing gas reservoirs; (**c**) the TPG characteristics of the startup pressure gradient.

In addition, when gas circulates in water-bearing rock samples, water molecules typically adhere to pore walls in the form of a water film or enter smaller pores due to the pressure exerted by the gas and capillaries. However, as gas molecules are constantly alternating within the porous rock system, they have difficulty colliding with solid pore walls and can only collide with water molecules. This limitation limits the occurrence of slip flow. As water saturation increases, more water molecules cover the pore walls, reducing the frequency of collisions between gas molecules and solid pore walls and further limiting slip flow. In particular, when water saturation reaches 40%, the gas slip effect is almost completely restricted. Therefore, when studying gas reservoirs with water saturations above 40%, the gas slip effect can be neglected.

(2) Stress sensitivity effect.

The analysis from Figure 12b leads to the following conclusion: an increase in water saturation results in a decrease in the slip factor, with changes in pore structure being the primary cause of this phenomenon. When water enters the rock framework, interfacial tension forms a water film, reducing the radii of pores and throats, leading to a decline in the formation's permeability and increasing the stress sensitivity of the reservoir. The presence of water reduces the compressive strength of the rock framework, causing further compression of particles and pores. Under high water saturation conditions, water forms a film on the surface of quartz particles. When effective stress increases, the water film loses balance, leading to pressure dissolution and $SiO_2$ precipitation. Consequently, this reduces the radii of pores and throats as well as the permeability.

When the water saturation is less than 40%, the influence of water saturation on the stress sensitivity coefficient is minimal and can almost be neglected. However, when the water saturation exceeds 40%, the impact of water saturation on the stress sensitivity coefficient exhibits an exponential increase trend. Therefore, when studying gas reservoirs with water saturation exceeding 40%, it is crucial to consider the significance of water saturation's effect on stress sensitivity.

(3) Threshold pressure gradient

Based on the analysis from Figure 12c, the following conclusions can be drawn: when the water saturation is less than 30%, no TPG occurs during gas penetration. However, when the water saturation exceeds 40%, the TPG significantly changes with the increase in water saturation. This phenomenon is due to the fact that at higher water saturations, gas in the reservoir cannot exist in a continuous phase but disperses into multiple small bubbles for flow. As these small bubbles pass through the throats, they induce the Jamin effect at each throat, causing capillary pressures to add up in the displacement direction. On a macroscopic scale, the higher the reservoir's water saturation, the larger the value of the TPG, as this additive effect is more likely to occur in throats with thicker water films.

In summary, once the water saturation in tight gas reservoirs exceeds 40%, there is a significant change in the seepage mechanisms. The gas slip flow effect no longer plays a role, and stress sensitivity effects and the TPG become significantly influenced by water saturation. Therefore, it is necessary to categorize tight sandstone gas reservoirs with water saturation exceeding 40% separately. These can be referred to as "High-Water-Cut Tight Gas Reservoir", aiming for a better understanding and description of their seepage characteristics and behaviors.

### 3.2. Zoning Productivity Physicochemical Mode of Fractured Horizontal Wells in High-Water-Cut Tight Gas Reservoirs under Complex Seepage Conditions

Based on the performance of fractured gas wells in different production stages and the characteristics of gas flow during each stage, the fluid flow region from the reservoir to the fractured horizontal well can be divided into three distinct zones (As shown in Figure 13). These three zones include Zone I: a high-speed non-Darcy flow region inside the fractures. In this zone, gas flows at high speed in the fractures with predominant features being high-speed non-Darcy effects and stress-sensitive effects within the fractures. It is noteworthy

that, unlike in vertical fractured wells, the cross-sectional area of horizontal fractured wells is much larger. This leads to differences in fluid seepage compared to vertical fractures, such as radial flow near the wellbore due to the convergence effect after the fluid linearly enters the fractures. Zone II: an elliptical non-Darcy flow region caused by boundary leakage at the fracture boundaries. In this zone, leakage at the fracture boundaries results in an elliptical non-Darcy flow region, where stress-sensitive effects in the matrix and the TPG effect on gas flow dominate. Zone III: an elliptical low-speed non-Darcy seepage region formed by fluid inflow from the surrounding formations. In this zone, fluid inflow from the surrounding formations creates an elliptical low-speed non-Darcy seepage region. The main characteristics of this zone are the simultaneous presence of stress-sensitive effects and TPG effects.

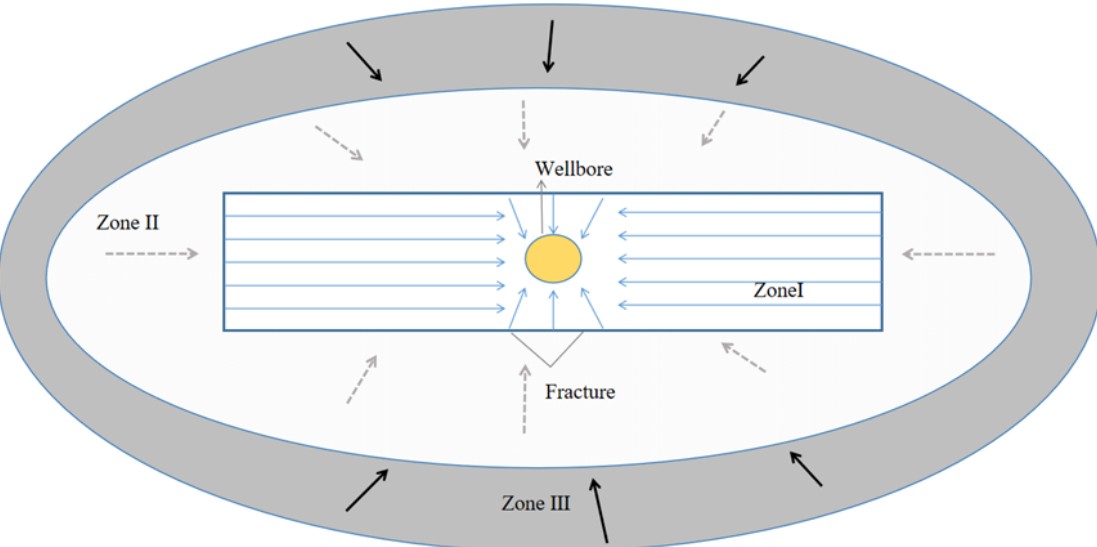

**Figure 13.** Fractured horizontal well planar flow schematic diagram.

This zoning helps to better understand the gas flow characteristics in different regions of the gas wells and the impact of various seepage effects. This information is crucial for optimizing gas well production and reservoir management.

When establishing seepage models, the reservoir and fractures are often regarded as two independent permeable systems that mutually influence each other. They are connected based on the principle of equal seepage rates and pressures between the reservoir and fractures. This connection condition not only effectively describes the interaction between the reservoir and fractures but also enhances the accuracy and reliability of the model. Therefore, during the model design, it is essential to consider the mutual influence between the reservoir and fractures to better simulate real-world situations.

Basic Assumptions of the Mathematical Model:

1. Perforated and fractured horizontal wells are located in a central position of an isotropic, isothermal, top and bottom-sealed gas reservoir. Only the heterogeneity of permeability is considered.
2. These artificial fractures are evenly distributed vertically along the wellbore, mirroring the gas reservoir's height. They exhibit symmetrical distribution around the wellbore and possess infinite conductivity, situated at the midpoint of the gas reservoir.
3. Isothermal gas seepage occurs, following the high-speed non-Darcy flow law inside the artificially fractured fractures. Fluid flow between matrix and fracture systems adheres to the low-speed non-Darcy seepage law.
4. Both the reservoir and gas are compressible. The influence of gravity and capillary forces is neglected.

5. The steady-state seepage of homogeneous gas–water two-phase fluids is assumed. Temperature changes and special physicochemical phenomena are ignored.

6. Contamination on the fracture walls is not considered, and contamination in the near-wellbore zone and the reservoir is neglected. The effects of gravity and capillary forces are ignored, as well as the pressure drop in the wellbore.

Based on the given assumptions, the physical model of the fractured horizontal well is formulated as shown in Figure 14. This model is designed for tight sandstone gas reservoirs. In the model, the horizontal well has a length of L and a radius of $r_w$. Additionally, the model includes $N$ fractures with a half-length of $L_f$, each fracture has a width of $W_f$, the drainage radius is $r_e$, and the gas layer thickness of h. The model is situated within the heterogeneous space O-XYZ, where the primary permeability is oriented along the X, Y, and Z directions. The main permeability aligned with the $X$, $Y$, and $Z$ directions are denoted as $K_x$, $K_y$, and $K_z$, respectively. $K_z$ points in the vertical direction, the angle between the horizontal wellbore and the primary permeability $K_x$ is θ, and the fractures are perpendicular to the wellbore with permeability of $K_{fi}$.

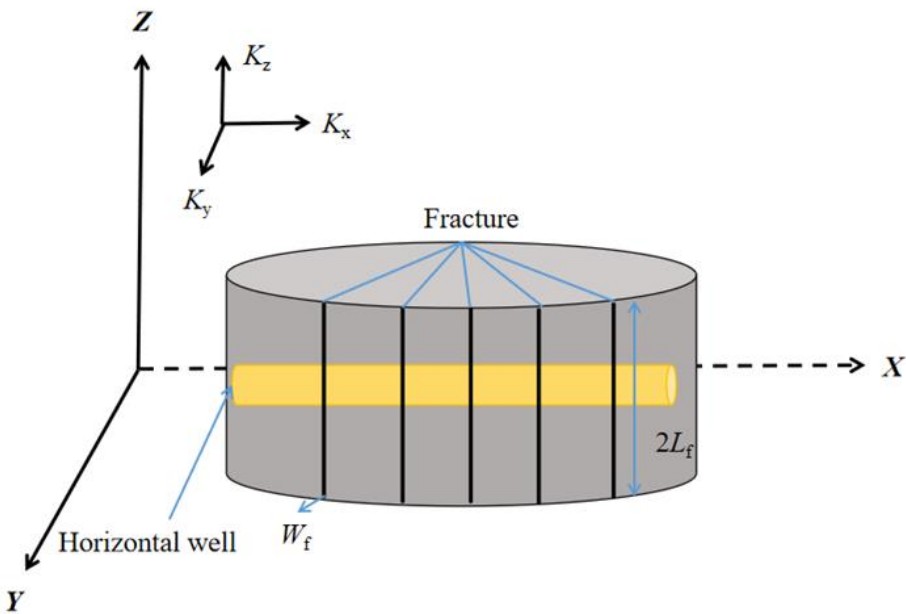

**Figure 14.** Fractured well permeability model.

*3.3. Transformation of Heterogeneity*

In the heterogeneous space *O-XYZ*, coordinate transformation is used to equivalently transform a permeability heterogeneous gas reservoir into a homogeneous gas reservoir.

$$\begin{cases} \varepsilon_1 = \sqrt{\dfrac{K_y}{K_z}} \\ \varepsilon_2 = \sqrt{\dfrac{K_z}{K_x}} \\ \varepsilon_3 = \sqrt{\dfrac{K_x}{K_y}} \\ K = \sqrt[3]{K_x K_y K_z} \end{cases} \tag{9}$$

The following non-uniform coordinate transformation is performed:

$$\begin{cases} X_1 = X\sqrt{\dfrac{K}{K_x}} \\ Y_1 = Y\sqrt{\dfrac{K}{K_y}} \\ Z_1 = Z\sqrt{\dfrac{K}{K_z}} \\ K_1 = K \end{cases} \tag{10}$$

After coordinate transformation, the seepage problem in the heterogeneous space *O-XYZ* is transformed into a problem in the homogeneous space $O\text{-}X_1Y_1Z_1$. At this point, in space $O\text{-}X_1Y_1Z_1$, the average permeability is $K_1$, the half-length of the fracture is $L_{f1}$, the width of the fracture is $W_{f_1}$, the gas release radius is $r_{el}$, the gas reservoir thickness is $h_1$, and the wellbore radius is $r_{wl}$.

$$L_{f1} = L_f \left(\frac{\varepsilon_2}{\varepsilon_3}\right)^{1/3} \tag{11}$$

$$h_1 = h \left(\frac{\varepsilon_1}{\varepsilon_2}\right)^{1/3} \tag{12}$$

$$W_{f1} = W_f \left(\frac{\varepsilon_3}{\varepsilon_1}\right)^{1/3} \tag{13}$$

$$r_{el} = \frac{r_e}{2} \left( \left(\frac{\varepsilon_2}{\varepsilon_3}\right)^{1/3} + \left(\frac{\varepsilon_3}{\varepsilon_1}\right)^{1/3} \right) \tag{14}$$

After a coordinate transformation, the horizontal cross-section of the hydraulic fracturing wellbore in the *O-XYZ* space has transitioned from a circle with a radius of $r_w$ to an ellipse within the $O\text{-}X_1Y_1Z_1$ space, characterized by two semi-axes $r_w(\varepsilon_2/\varepsilon_3)^{1/3}$ and $r_w(\varepsilon_3/\varepsilon_1)^{1/3}$. When the fluids near the wellbore reach a state of equilibrium, the wellbore radius is equal to the arithmetic mean of the major and minor semi-axes. Therefore, the wellbore diameter of the hydraulic fracturing well after the coordinate transformation is:

$$r_{wl} = \frac{r_w}{2} \left( (\varepsilon_2/\varepsilon_3)^{1/3} + (\varepsilon_2/\varepsilon_3)^{1/3} \right) \tag{15}$$

*3.4. Derivation of Zoning Productivity Calculation Method for Fractured Horizontal Wells*

The production of tight high-water gas wells is divided into three zones: Zone I represents the high-speed non-Darcy flow in fractures, Zone II reflects the elliptical Darcy flow controlled by fracture boundaries, and Zone III represents the low-permeability non-Darcy flow in the matrix. To establish a production forecasting model for fractured gas wells, the concept of perturbed ellipses and the idea of equivalent developing rectangles are introduced. This model comprehensively considers the effects of gas slippage, TPG, and stress sensitivity on productivity, providing a theoretical basis for the rational determination of productivity in high-water-cut tight gas reservoirs.

3.4.1. Zone I Productivity Prediction Model

In the fracturing operation of horizontal gas wells, the fracturing stage is significant and crucial. During this crucial period, the permeability in the fractures is usually at a relatively high level, enabling gas to flow within the fractures at high speeds through non-Darcy flow. Additionally, due to the significantly larger cross-sectional area of the transverse fractures in fractured horizontal wells compared to the cross-sectional area of the wellbore, the fluid flow process differs from that in vertical wells with vertical fractures. After linear flow into the fractures, fluid flow near the wellbore transitions to radial flow due to the confluence effect. In this complex process, the high-speed non-Darcy effect and stress-sensitive effect are very significant. It is worth noting that the characteristics of this region allow us to temporarily exclude the effects of the TPG and slippage.

(1)    Linear flow stage.

In the fluid movement within the fracture, from the tip of the fracture to the point where the linear and radial flows intersect, the flow conditions can be equivalent to the flow in a strip-shaped formation with a length of $L_{f1}$, a width of $W_{f1}$, and a height of $h_1$. On this basis, assuming that the radius at the intersection of linear and radial flows within the fracture is $r_x$, the corresponding pressure is $p_x$.

The equations governing the movement of gas and water phases within the fracture are:

Gas phase:

$$\begin{cases} \frac{dp}{dx} = \frac{1}{8.64\times10^{-2}} \frac{\mu_g v_g}{K_f K_{rg}} + \frac{1}{8.64\times10^{11}} \beta_f \rho_g v_g^2 \\ v_g = 5\times10^3 \frac{\rho_{gsc} q_{gsc}}{2\rho_g w_f h} \end{cases} \tag{16}$$

Water phase:

$$\begin{cases} \frac{dp}{dx} = \frac{1}{8.64\times10^{-2}} \frac{\mu_w v_w}{K_f K_{rw}} \\ v_w = 5\times10^3 \frac{\rho_{wsc} q_{wsc}}{2\rho_w w_f h} \end{cases} \tag{17}$$

where $r$ is the linear seepage distance along the centerline of the fracture (m); $K_f$ is the absolute permeability of the fracture (mD); $\beta_f$ is the Forchheimer coefficient (m$^{-1}$); $v_g$ is the gas velocity within the fracture (m/d); $v_w$ is the water phase velocity within the fracture (m/d); and $K_{rg}$ is the relative permeability of the gas phase under two-phase gas–water seepage conditions within the fracture. $K_{rw}$ is the relative permeability of the water phase under two-phase gas–water seepage conditions within the fracture. $w_{f1}$ is the width of the fracture (m). $h_1$ is the height of the fracture (m).

Where $\beta_f = \frac{m}{K_{fi}^n}$, under conditions where different proppants are used, m and n values are as shown in Table 6.

**Table 6.** Under different proppants, m and n value table.

| Proppant Size (mm) | n | m |
|---|---|---|
| 2.12~3.18 | 1.21 | 3.23 |
| 1.27~2.54 | 1.34 | 2.63 |
| 0.64~1.27 | 1.65 | 1.65 |
| 0.42~0.64 | 1.60 | 1.10 |

Considering the stress-sensitive effect of water saturation:

$$\begin{cases} K_f = K_{fi}\left(\frac{p_c - p_p}{p_c - p_i}\right)^{-s_1} \\ S_1 = aK^{-b} \\ a = 0.28421 + 0.000148694 * e^{(S_w/0.09258)} \\ b = -0.0317 * e^{(S_w/0.17024)} + 0.31759 \end{cases} \tag{18}$$

where $K_f$ represents the initial permeability of the fracture (mD); $a$ is the coefficient parameter for the stress sensitivity of water saturation; and $b$ is the exponent parameter for the stress sensitivity of water saturation.

Considering the stress sensitivity and turbulent effects during the linear flow stage of gas in the fracture, the flow equation for hydraulic fracturing horizontal wells can be derived as follows:

Gas phase:

$$\begin{cases} \frac{dp}{dx} = \frac{1}{8.64\times10^{-2}} \frac{\mu_g v_g}{K_{fi}\left(\frac{p_c - p_p}{p_c - p_i}\right)^{-s_1} K_{rg}} + \frac{1}{8.64\times10^{11}} \beta_f \rho_g v_g^2 \\ v_g = 5\times10^3 \frac{\rho_{gsc} q_{gsc}}{2\rho_g w_f h} \end{cases} \tag{19}$$

Water phase:

$$\begin{cases} \frac{dp}{dx} = \frac{1}{8.64\times10^{-2}} \frac{\mu_w v_w}{K_{fi}\left(\frac{p_c - p_p}{p_c - p_i}\right)^{-s_1} K_{rw}} \\ v_w = 5\times10^3 \frac{\rho_{wsc} q_{wsc}}{2\rho_w w_f h} \end{cases} \tag{20}$$

The following equations can be obtained by combining Equations (17) and (18):

$$
\begin{aligned}
&(\rho_w \tfrac{K_{rw}}{\mu_w} + \rho_g \tfrac{K_{rg}}{\mu_g}) \mathrm{K}_{fi} \left( \tfrac{p_c - p_p}{p_c - p_i} \right)^{-s_1} dp = \\
&(5.787 \tfrac{\rho_w q_w + \rho_{gsc} q_{gsc}}{\mathrm{K}_{fi} \left( \tfrac{p_c - p_p}{p_c - p_i} \right)^{-s_1} w_f h} + 3.35 \times 10^{-6} \beta_f \tfrac{K_{rg}}{\mu_g} \tfrac{\rho_g^2 q_g^2}{w_f^2 h^2} \mathrm{K}_{fi} \left( \tfrac{p_c - p_p}{p_c - p_i} \right)^{-s_1}) dx
\end{aligned}
\tag{21}
$$

The definition of two-phase pseudo-pressure within the fracture is as follows:

$$
\varphi_1(p) = \int_0^p f(p) dp = (\rho_w \tfrac{K_{rw}}{\mu_w} + \rho_g \tfrac{K_{rg}}{\mu_g}) \mathrm{K}_{fi} \left( \tfrac{p_c - p_p}{p_c - p_i} \right)^{-s_1} dp
\tag{22}
$$

Integrating the equation resulting from the combination of Equations (19) and (20):

$$
\begin{aligned}
&\varphi_1(p_f) - \varphi_1(p_x) = 5.787 \times 10^4 \times \frac{\left(1 + \tfrac{\rho_w}{\rho_{gsc}} WGR\right) \rho_{gsc} (L_{f1} - r_x)}{\mathrm{K}_{fi} w_{f1} h_1} q_{gsc} + \\
&q_{qsc}^2 \int_{r_x}^{L_{f1}} \left[ 3.35 \times 10^{-6} \beta_f \tfrac{K_{rg}}{\mu_g} \tfrac{\rho_{gsc}^2 (x_f - r_w)}{4 w_{f1}^2 h_1^2} \left( \tfrac{p_c - p_p}{p_c - p_i} \right)^{-s_1} \right] dx
\end{aligned}
\tag{23}
$$

where $p_f$ is the pressure at the fracture tip (MPa); $p_x$ is the pressure at the boundary between the linear seepage field and the radial seepage field within the fracture (MPa); $r = h/2$; $r_x$ is the radial radius at the boundary between the linear seepage field and the radial seepage field within the fracture (m).

(2) Radial flow stage

When the fluid in the fracture starts to undergo radial flow, it is equivalent to flowing from the boundary to the wellbore with radius $r_{w1}$ in a circular formation with supply radius $r_x$, supply boundary pressure $p_x$, and formation thickness $W_{f1}$. Based on the previous derivation, the following seepage equations of motion can be obtained for the gas and water phases in a single fracture of a fractured horizontal well during the radial flow stage in the fracture, respectively, under the consideration of the high-velocity non-Darcy and stress-sensitive effects.

Gas phase:

$$
\begin{cases}
\dfrac{dp}{dr} = \dfrac{1}{8.64 \times 10^{-2}} \dfrac{\mu_g v_g}{K_f K_{rg}} + \dfrac{1}{8.64 \times 10^{11}} \beta_f \rho_g v_g^2 \\
v_g = 5 \times 10^3 \dfrac{\rho_{gsc} q_{gsc}}{2\pi r w_{f1} \rho_g} \\
\mathrm{K}_f = \mathrm{K}_{fi} \left( \dfrac{p_c - p_p}{p_c - p_i} \right)^{-s_1}
\end{cases}
\tag{24}
$$

Water phase:

$$
\begin{cases}
\dfrac{dp}{dr} = \dfrac{1}{8.64 \times 10^{-2}} \dfrac{\mu_w v_w}{K_f K_{rw}} \\
v_w = 5 \times 10^3 \dfrac{\rho_{wsc} q_{wsc}}{2\pi r w_{f1} \rho_w} \\
\mathrm{K}_f = \mathrm{K}_{fi} \left( \dfrac{p_c - p_p}{p_c - p_i} \right)^{-s_1}
\end{cases}
\tag{25}
$$

The following equations can be obtained by combining Equations (22) and (23):

$$
\begin{aligned}
&(\rho_w \tfrac{K_{rw}}{\mu_w} + \rho_g \tfrac{K_{rg}}{\mu_g}) \left( \tfrac{p_c - p_p}{p_c - p_i} \right)^{-s_1} dp = \\
&(5.787 \tfrac{\rho_w q_w + \rho_{gsc} q_{gsc}}{\pi r w_{f1}} + 3.35 \times 10^{-6} \beta_f \tfrac{K_{rg}}{\mu_g} \tfrac{\left( \tfrac{p_c - p_p}{p_c - p_i} \right)^{-s_1} \rho_{gsc}^2 q_{gsc}^2}{(\pi r w_{f1})^2}) dr
\end{aligned}
\tag{26}
$$

The pseudo-pressure for the two phases within the fracture is defined. Combining Equations (20) and (22), and integrating from the boundary of the radial seepage field within the fracture (radius $r_x$) to the wellbore, the productivity equation for any single fracture in the radial seepage stage of the fracture, considering stress sensitivity and high-

speed non-Darcy effects, can be obtained under steady-state gas–water flow conditions.

$$\varphi_1(p_x) - \varphi_1(p_{wf}) = 5.787 \times 10^4 \times \frac{\left(1 + \frac{\rho_w}{\rho_{gsc}} WGR\right)\rho_{gsc}}{K_{fi}w_{fl}h} \ln(\frac{r_x}{r_{w1}})q_{gsc} + $$
$$q_{qsc}^2 \int_{r_{w1}}^{r_x} \left[3.35 \times 10^{-6} \beta_f \frac{K_{rg}}{\mu_g} \frac{\rho_{gsc}^2 \left(\frac{p_c - p_p}{p_c - p_i}\right)^{-s_1}}{(2\pi w_{f1}r)^2}\right] dr \tag{27}$$

Due to the principle of water and electricity similitude, the fluid flow rates in the two seepage fields are equal. Therefore, by simultaneously considering Equations (21) and (25), a productivity prediction model for a single fracture in hydraulic fracturing horizontal wells in tight sandstone gas reservoirs is derived. This model takes into account permeability heterogeneity, high-speed non-Darcy flow, gas slippage phenomenon, and stress sensitivity.

$$\begin{cases} \varphi_1(p_f) - \varphi_1(p_x) = 5.787 \times 10^4 \times \frac{\left(1 + \frac{\rho_w}{\rho_{gsc}} WGR\right)\rho_{gsc}(L_{f1} - r_x)}{K_{fi}w_{f1}h_1}q_{gsc} + \\ q_{qsc}^2 \int_{r_x}^{L_{f1}} \left[3.35 \times 10^{-6} \beta_f \frac{K_{rg}}{\mu_g} \frac{\rho_{gsc}^2(x_f - r_w)}{4w_{f1}^2 h_1^2} \left(\frac{p_c - p_p}{p_c - p_i}\right)^{-s_1}\right] dx \\ \varphi_1(p_x) - \varphi_1(p_{wf}) = 5.787 \times 10^4 \times \frac{\left(1 + \frac{\rho_w}{\rho_{gsc}} WGR\right)\rho_{gsc}}{K_{fi}w_{fl}h} \ln(\frac{r_x}{r_{w1}})q_{gsc} + \\ q_{qsc}^2 \int_{r_{w1}}^{r_x} \left[3.35 \times 10^{-6} \beta_f \frac{K_{rg}}{\mu_g} \frac{\rho_{gsc}^2 \left(\frac{p_c - p_p}{p_c - p_i}\right)^{-s_1}}{(2\pi w_{f1}r)^2}\right] dr \end{cases} \tag{28}$$

### 3.4.2. Region II Productivity Prediction Model

In Region II of horizontal gas wells, the fracture boundaries induce a planar two-dimensional elliptical seepage phenomenon within the formation (As shown in Figure 15). In this scenario, gas seepage is influenced by the TPG, and stress-sensitive effects become apparent. It is worth mentioning that the influence of slip flow can be neglected. Gas seepage under these conditions conforms to the characteristics of low-speed non-Darcy elliptical seepage.

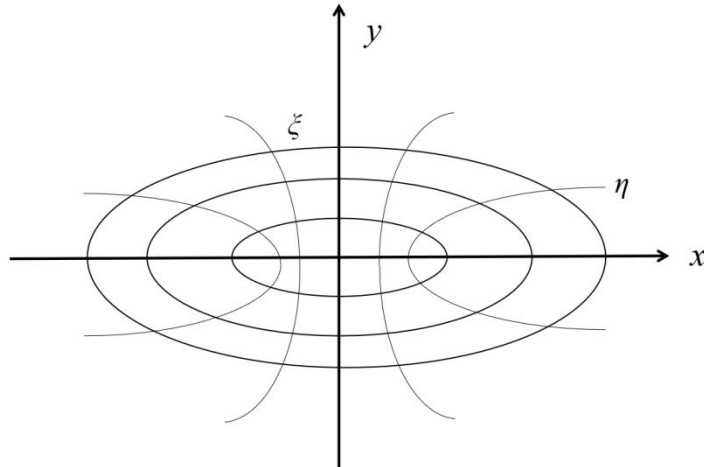

**Figure 15.** Conceptual diagram of the elliptical flow field within the formation.

In the region around the horizontally fractured well, the external seepage field shows a series of elliptical curves (see Figure 14). When the formation is in the non-Darcy flow zone, the seepage form is elliptical seepage, and the elliptical seepage law can be applied to

describe its seepage characteristics. There is the following relationship between elliptical coordinates and right-angle coordinates:

$$\begin{cases} x = a\cos\eta \\ y = b\sin\eta \\ a = x_f\cosh\xi \\ b = x_f\sinh\xi \\ \xi = \sinh^{-1}\left(\frac{2\pi r}{x_f}\right) \end{cases} \tag{29}$$

where $a$ is the half-axis length of the elliptical seepage field, m; $b$ is the short half-axis length of the elliptical seepage field, m.

The seepage field of elliptical seepage is usually described by a developing family of rectangles, so the average short half-axis is:

$$\bar{y} = \frac{2}{\pi}\int_1^{\frac{\pi}{2}} y d\eta = \frac{2b}{\pi} = \frac{2x_f\sinh\xi}{\pi} \tag{30}$$

The pressure gradient in the elliptical coordinate system is:

$$\frac{dp}{dy} = \frac{dp}{d\xi}\frac{d\xi}{dy} = \frac{\pi}{2x_f\cosh\xi}\frac{dp}{d\xi} \tag{31}$$

In gas–water two-phase percolation, the continuity equations are as follows:
Gas phase:

$$\begin{cases} \frac{dp}{dr} = \frac{\mu_g v_g}{8.64\times10^{-2}K_g K_{rg}} \\ v_g = 2500\frac{\rho_{gsc}q_{gsc}}{\rho_g x_f\cosh\bar{\xi}} \end{cases} \tag{32}$$

Water phase:

$$\begin{cases} \frac{dp}{dr} = \frac{\mu_w v_w}{8.64\times10^{-2}aK_{wi}e^{bp}K_{rw}} \\ v_w = 2500\frac{\rho_w q_w}{\rho_w x_f\cosh\bar{\xi}} \end{cases} \tag{33}$$

where $p$ is the formation pressure, MPa; $r$ is the radial radius, m; $m_r$ is gas mass flow at radius r, kg/s; $m_w$ is the gas mass flow at radius r, kg/s; $\mu_g$ is the viscosity of gas, MPa·s; $\mu_g$ is the viscosity of water, MPa·s; $K_{rg}$ is the gas phase relative permeability, $10^{-3}$ μm³; and $K_{rw}$ is the relative permeability of water phase, $10^{-3}$ μm³.

Considering the influence of water saturation on stress sensitivity and TPG, the permeability is revised as follows:
Gas phase:

$$\begin{cases} \frac{dp}{dr} - \lambda_g = \frac{\mu_g v_g}{8.64\times10^{-2}K_{fi}\left(\frac{p_c-pp}{p_c-p_i}\right)^{-s_1}K_{rg}} \\ v_g = 2500\frac{\rho_{gsc}q_{gsc}}{\rho_g l_f\cosh\bar{\xi}} \\ \lambda_g = ak^{(-b)} = \left(1.41741*10^{-6}*e^{(S_w/5.36345)}\right)K_{fi}^{-\left(86.25654*S_w^{(-0.91146)}\right)} \end{cases} \tag{34}$$

Water phase:

$$\begin{cases} \frac{dp}{dr} - \lambda_w = \frac{\mu_w v_w}{8.64\times10^{-2}K_{fi}\left(\frac{p_c-pp}{p_c-p_i}\right)^{-s_1}K_{rw}} \\ v_w = 2500\frac{\rho_w q_w}{\rho_w l_f\cosh\bar{\xi}} \end{cases} \tag{35}$$

The following equations can be obtained by combining Equations (32) and (33):

$$
\begin{aligned}
&\left(\rho_g \frac{K_{rg}}{\mu_g} + \rho w \frac{K_{rw}}{\mu_w}\right)K_{fi}\left(\frac{p_c - p_p}{p_c - p_i}\right)^{-s_1} dp = \\
&\left(\begin{array}{l} \frac{11.574 \times (\rho_{gsc}q_{gsc} + \rho_{wsc}q_{wsc})}{2\pi r h_1 K_1} + \\ \left(\left(\rho_g \frac{K_{rg}}{\mu_g}\right)\lambda_g + \left(\rho_w \frac{K_{rw}}{\mu_w}\right)\lambda_w\right)K_{fi}\left(\frac{p_c - p_p}{p_c - p_i}\right)^{-s_1} \cosh\xi d\xi \end{array}\right) d\xi
\end{aligned}
\tag{36}
$$

The pseudo-pressure of two-phase formation is defined as:

$$
\varphi_2(p) = \int_0^p f(p)dp = \int_0^p \left(\rho_g \frac{K_{rg}}{\mu_g} + \rho w \frac{K_{rw}}{\mu_w}\right)K_{fi}\left(\frac{p_c - p_p}{p_c - p_i}\right)^{-s_1} dp
\tag{37}
$$

The pseudo-startup pressure of two phases is defined as:

$$
\lambda_{gw}(p) = \left(\left(\rho_g \frac{K_{rg}}{\mu_g}\right)\lambda_g + \left(\rho_w \frac{K_{rw}}{\mu_w}\right)\lambda_w\right)K_{fi}\left(\frac{p_c - p_p}{p_c - p_i}\right)^{-s_1}
\tag{38}
$$

Integrating the equation resulting from the combination of Equations (34)–(36):

$$
\begin{aligned}
&\varphi_2(p_\xi) - \varphi_2(p_f) = \left(\frac{11.574 \times \left(1 + \frac{\rho_{wsc}}{\rho_{gsc}}WGR\right)}{2\pi h_1 K_1} \ln\left(\frac{a + \sqrt{a^2 - x_f^2}}{x_f}\right)\right)q_{sc} \\
&+0.637 L_{fl}\int_{\xi_f}^{\xi} \lambda_{gw}d\xi
\end{aligned}
\tag{39}
$$

where $a$ is the semi-major axis length of the hydraulic fracture flow ellipse (m); $\xi$ corresponds to an ellipse with a long axis of $r_{el}$ and $\xi_f$ corresponds to an ellipse with an average short axis of $W_{f1}/2$; $p_f$ is the fracture tip pressure, MPa, and $\xi$ is an elliptic coordinate; $\xi_e$ is the coordinates of the gas reservoir supply boundary in the elliptic coordinate system, dimensionless; and $\xi_f$ is the coordinates of the fracture tip in the elliptic coordinate system, dimensionless.

### 3.4.3. Region III Productivity Prediction Model

In Region III, due to the continuous depletion of reservoir energy, fractures deform, resulting in relatively low permeability. During this stage, gas production mainly relies on the release of gas from the formation matrix in the far-field area under low-pressure conditions. It is important to emphasize that this study focuses on tight water-bearing gas reservoirs. Therefore, the effect of slippage is no longer considered. Moreover, during this process, stress sensitivity gradually decreases, the effect of the TPG impacts gas seepage, and it complies with the laws of low-speed non-Darcy flow.

Referring to the seepage equations in Section 3.4.2, the gas–water biphasic productivity equation for a single fracture in hydraulic fracturing horizontal wells, considering the TPG and stress-sensitive effects under stable seepage conditions in the formation, can be derived as follows:

$$
\begin{aligned}
&\varphi_2(p_e) - \varphi_2(p_\xi) = \left(\frac{11.574 \times \left(1 + \frac{\rho_{wsc}}{\rho_{gsc}}WGR\right)}{2\pi h_1 K_1} \ln\left(\frac{r_e}{r_\xi}\right)\right)q_{sc} \\
&+0.637 L_{fl}\int_{\xi}^{\xi_e} \lambda_{gw}d\xi
\end{aligned}
\tag{40}
$$

where $r_e$ is the supply radius (m); $p_\xi$ is the pressure at the interface (MPa); and $r_\xi$ is the distance from the well axis at the interface (m).

### 3.4.4. Establishment of Hydraulic Fracturing Horizontal Well Productivity Equation

(1)　Single Fracture Productivity Equation Development

The production of fluids from the formation results from the series connection of the seepage field within the formation and the linear seepage field within the fractures. When all three flow regions are simultaneously seeping, according to the principle of mass conservation, the seepage rates and pressures at the interfaces between adjacent

flow regions should be equal. This allows the elimination of the interface pressures. By simultaneously solving Equations (26), (37) and (38), a hydraulic fracturing horizontal well prediction model is obtained, considering factors such as non-Darcy flow at low and high speeds, stress sensitivity, and TPG. This model can be used to predict the productivity of tight, highly saturated gas reservoirs.

$$
\begin{cases}
\varphi_1(p_f) - \varphi_1(p_x) = 5.787 \times 10^4 \times \dfrac{\left(1 + \frac{\rho_w}{\rho_{gsc}} WGR\right)\rho_{gsc}(L_{f1}-r_x)}{K_{fi}w_{f1}h_1} q_{gsc} + \\
q_{qsc}^2 \int_{r_x}^{L_{f1}} \left[3.35 \times 10^{-6}\beta_f \dfrac{K_{rg}}{\mu_g} \dfrac{\rho_{gsc}^2(x_f-r_w)}{4w_{f1}^2 h_1^2}\left(\dfrac{p_c-p_p}{p_c-p_i}\right)^{-s_1}\right]dx \\[2mm]
\varphi_1(p_x) - \varphi_1(p_{wf}) = 5.787 \times 10^4 \times \dfrac{\left(1 + \frac{\rho_w}{\rho_{gsc}} WGR\right)\rho_{gsc}}{K_{fi}w_{fl}h}\ln\!\left(\dfrac{r_x}{r_{w1}}\right)q_{gsc} + \\
q_{qsc}^2 \int_{r_{w1}}^{r_x}\left[3.35 \times 10^{-6}\beta_f \dfrac{K_{rg}}{\mu_g} \dfrac{\rho_{gsc}^2\left(\frac{p_c-p_p}{p_c-p_i}\right)^{-s_1}}{(2\pi w_{f1}r)^2}\right]dr \\[2mm]
\varphi_2(p_\zeta) - \varphi_2(p_f) = \left(\dfrac{11.574\times\left(1+\frac{\rho_{wsc}}{\rho_{gsc}}WGR\right)}{2\pi h_1 K_1}\ln\!\left(\dfrac{a+\sqrt{a^2-x_f^2}}{x_f}\right)\right)q_{sc} \\
+0.637 L_{fl}\int_{\zeta_f}^{\zeta}\lambda_{gw}d\zeta \\[2mm]
\varphi_2(p_e) - \varphi_2(p_e) = \left(\dfrac{11.574\times\left(1+\frac{\rho_{wsc}}{\rho_{gsc}}WGR\right)}{2\pi h_1 K_1}\ln\!\left(\dfrac{r_e}{r_\zeta}\right)\right)q_{sc} \\
+0.637 L_{fl}\int_{\zeta}^{\zeta_e}\lambda_{gw}d\zeta
\end{cases}
\tag{41}
$$

(2)  Equivalent wellbore radius model

Under steady-state seepage conditions, if a productivity calculation model for a certain type of gas well, influenced by complex factors, is known, it can be compared with a general vertical well productivity calculation model that does not consider those complex factors under the general Darcy conditions. When the production rates of the two models are equal, the calculated equivalent radius of the general vertical well is referred to as the equivalent wellbore radius, denoted as $r_{\text{equ}}$, and is expressed as follows:

$$
\begin{aligned}
\varphi_1(p_e) - \varphi_1(p_{\text{wf}}) &= \left(\dfrac{11.574\times\left(1+\frac{\rho_{wsc}}{\rho_{gsc}}WGR\right)}{2\pi h_1 K_1}\ln\!\left(\dfrac{r_{e1}}{r_{\zeta\text{equ}}}\right)\right)q_{sc} \\
&\quad +\lambda_{gw}\overline{(p)}(r_{e1}-r_{equ})
\end{aligned}
\tag{42}
$$

(3)  Establishment of productivity equation for N fractures in hydraulic fracturing horizontal wells.

Building on any single fracture productivity calculation model derived from the previous text, employing the equivalent wellbore diameter theory, the single fracture is transformed into a regular vertical well. Accounting for inter-fracture interference, according to the pressure superposition principle, it can be expressed as:

$$
\begin{cases}
m_1(pe) - m_1(p_{wf1}) = \Delta m_1(p)_{11}q_{sc1} + \Delta m_1(p)_{21}q_{gsc2} + \ldots \Delta m_1(p)_{n1}q_{gscn} \\
m_1(pe) - m_1(p_{wf2}) = \Delta m_1(p)_{12}q_{sc1} + \Delta m_1(p)_{22}q_{gsc2} + \ldots \Delta m_1(p)_{n2}q_{gscn} \\
\quad\\
m_1(pe) - m_1(p_{wfn}) = \Delta m_1(p)_{1n}q_{sc1} + \Delta m_1(p)_{2n}q_{gsc2} + \ldots \Delta m_1(p)_{nn}q_{gscn}
\end{cases}
\tag{43}
$$

By solving the equations simultaneously, the production of N fractures in hydraulic fracturing horizontal wells can be obtained. The total production of hydraulic fracturing horizontal wells is:

$$
q_{gsc} = q_{gsc1} + q_{gsc2} + q_{gsc3} + \ldots q_{gscn} + \ldots q_{gscN}
\tag{44}
$$

### 3.5. Example Calculation

A horizontal hydraulic fracturing well in a tight, highly saturated gas reservoir undergoes a large-scale proppant fracturing transformation. The proppant used for hydraulic fracturing is a mixture of 70/140 mesh quartz sand and 40/70 mesh coated sand. The original formation pressure is 50 MPa, and the drainage radius of the reservoir is 200 m. The bottom hole flowing pressure is 25 MPa, the formation salinity is 1.06, and the reservoir thickness is 20 m. The density of water is 1 g/cm$^3$, with a reservoir water saturation of 65%. The viscosity of water is 1 mPa·s, and the formation temperature is 340 K. The initial formation pressure is 40 MPa, and the fracture width is 0.008 m. The effective half-length of the fracture is 106.2 m, and the absolute permeability of the fracture is 50 Darcy.

Using the model described in this paper and existing productivity calculation models, the productivity of the gas well is calculated separately, and the results are shown in the graph.

Figure 16 shows that the gas well productivity calculated using the zonal productivity model is significantly higher than the results obtained from existing models. This is because, compared to existing models, the model in this paper considers the seepage characteristics of the fracturing zone. After the fracturing transformation, the reservoir properties and seepage capacity in the transformed area significantly improved. As a result, the gas well productivity increases significantly compared to the traditional models that do not consider zonal divisions.

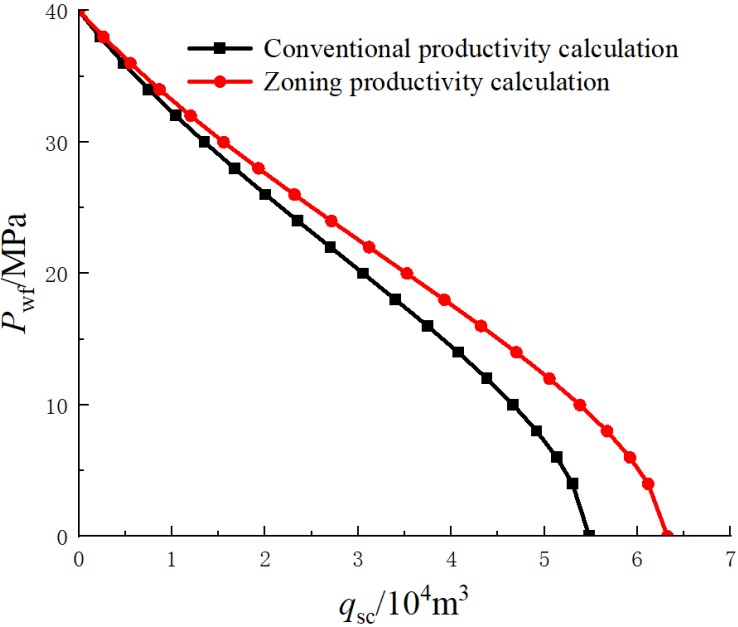

**Figure 16.** Comparison of results from different productivity calculation methods.

## 4. Sensitivity Analysis

Based on the validation from the practical example, a comparative analysis was conducted by adjusting different parameters in the model. This study aimed to investigate the impact of various factors on the zonal productivity calculation results.

### 4.1. The Impact of Differential Stress Sensitivity in Different Regions

Initially, by adjusting the stress sensitivity coefficients inside and outside the fracturing zone, the study explored the impact of differential stress sensitivity in different regions on the productivity of hydraulic fracturing gas wells in tight gas reservoirs. The results are shown in the figure.

The analysis of Figure 17 reveals that, considering zonal stress sensitivity, the productivity of the gas well significantly decreases compared to the scenario where stress

sensitivity is not considered. This decline is attributed to the continuous reduction in reservoir permeability as the development process progresses, significantly affecting the seepage capacity of fluids. Consequently, the productivity of the gas well experiences a significant decrease.

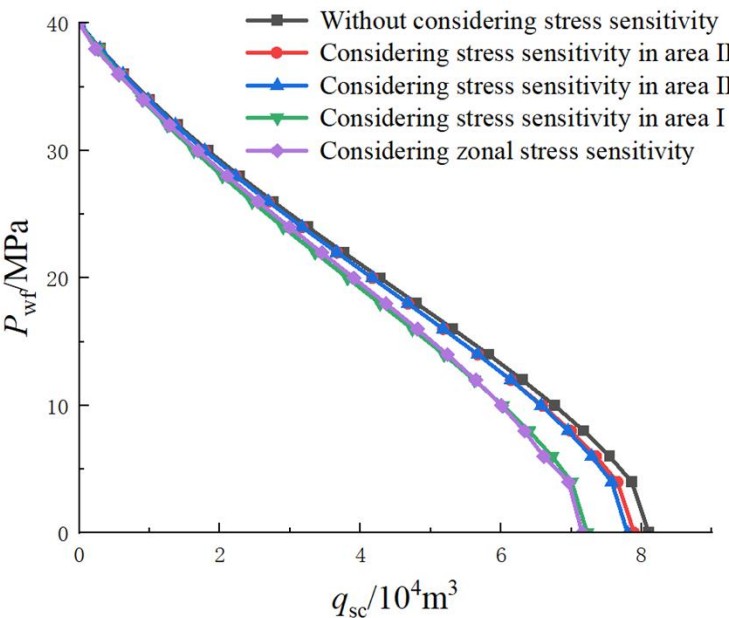

**Figure 17.** The impact of zonal stress sensitivity on the productivity of hydraulic fracturing gas wells in tight gas reservoirs.

The results in Figure 17 also indicate that, when considering stress sensitivity in a single region alone, the gas well productivity is lower compared to the scenario where stress sensitivity is not considered at all. Moreover, when stress sensitivity in Region I alone is considered, the gas well productivity is lower than when stress sensitivity in Regions II and III is taken into account. This is because although Region I has a higher sensitivity to stress and a more significant impact on seepage due to the presence of fractures, its area is much smaller than that of Regions II and III. Therefore, when stress sensitivity in Region I alone is considered, the productivity is lower than when stress sensitivity in Regions II and III is considered. In other words, only when the stress sensitivity characteristics of all three regions are considered simultaneously, the gas well productivity can be accurately determined.

### 4.2. The Impact of Dynamic TPG in Tight Reservoirs

The impact of dynamic TPG on the productivity of hydraulic fracturing gas wells in tight gas reservoirs is shown in Figure 18.

From Figure 18, it can be observed that considering the TPG results in a significant decrease in gas well productivity compared to not considering the TPG. This is because when the TPG is taken into account, the additional resistance to fluid flow increases, leading to a decrease in productivity. Furthermore, when considering the dynamic TPG, gas well productivity further decreases compared to the conventional fixed TPG. This is primarily because when accounting for the dynamic TPG, the TPG in the reservoir gradually increases as the development progresses. The lower the reservoir pressure, the larger the TPG, leading to a more pronounced decrease in productivity.

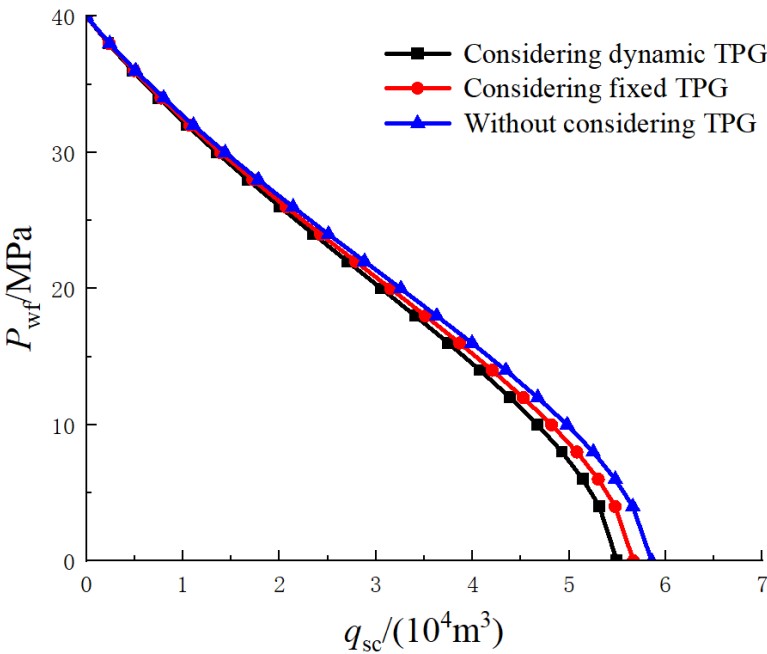

**Figure 18.** Effect of dynamic threshold pressure gradient on fractured gas well productivity in tight gas reservoirs.

### 4.3. The Impact of the Number of Fractures on Productivity

Analyzing Figure 19 shows that as the number of fractures increases, the IPR (Inflow Performance Relationship) curve shifts towards higher productivity. The gas well productivity increases, but the rate of increase becomes smaller. Figure 20 shows the variation in gas well productivity with the number of fractures. For the same hydraulic fracturing gas well, within a certain range, having more fractures leads to a larger effective fracture area, which is beneficial for increasing productivity. However, if the number of fractures continues to increase, the interference between fractures intensifies, leading to mutual inhibition among the fractures. As a result, the increase in gas well productivity becomes smaller, indicating that there exists an optimal number of fractures. Therefore, in practical production, it is crucial to optimize the number of fractures reasonably and avoid productivity losses caused by inter-fracture interference.

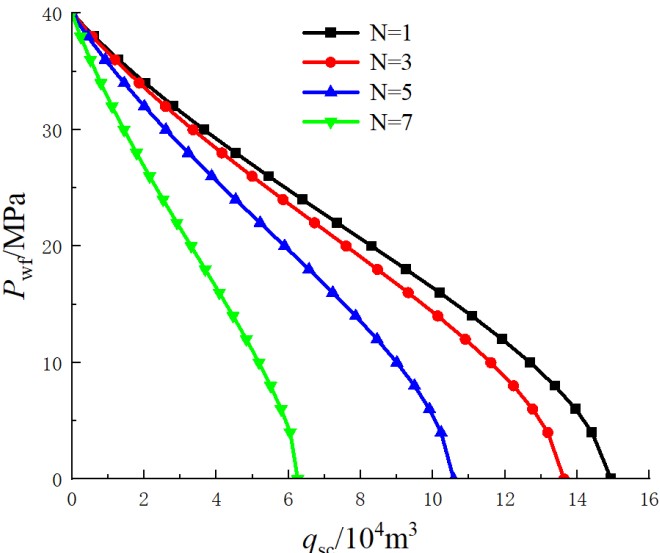

**Figure 19.** The impact of the number of fractures on gas well productivity.

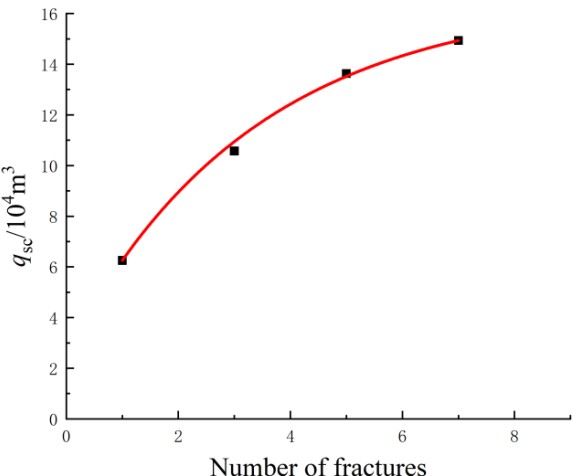

**Figure 20.** The curve depicting the variation in gas well productivity with the number of fractures.

*4.4. The Impact of Water–Gas Ratio (WRG) on Gas Well Productivity*

As shown in the Figure 21, different IPR curves are plotted under various water–gas ratios (WRGs): one without considering the stress sensitivity and threshold pressure gradient (TPGASS), one considering a fixed initiation pressure gradient and stress sensitivity, one considering the influence of water saturation on the TPG and stress sensitivity, and one showing the unobstructed flow rate variation curve with changes in the water–gas ratio considering the impact of water saturation on TPG and stress sensitivity. The analysis indicates that gas well productivity decreases after water production, and as the water–gas ratio increases, the curve shifts to the left. After water production, the water phase occupies the large pores initially occupied by the gas phase. More and more gas becomes discontinuous, reducing the relative permeability of the gas phase, which affects gas well productivity. With the increase in the water–gas ratio, the TPG and stress sensitivity effects increase, and the shift of the curve to the right becomes smaller. The lower the bottom hole flowing pressure, the greater the impact of TPG and stress sensitivity effects on gas well productivity. The variation curve of the production capacity with water-gas ratio is shown in Figure 22. Therefore, in productivity modeling, the influence of water saturation on the TPG and stress sensitivity effects should be considered. In practical production processes, appropriate operating procedures should be adopted to avoid early water breakthroughs or delay the time when the gas well starts producing water.

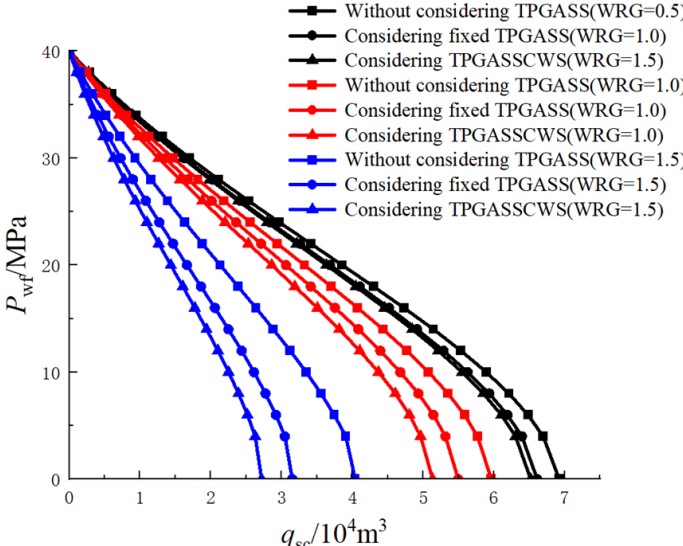

**Figure 21.** IPR curves under different WRGs.

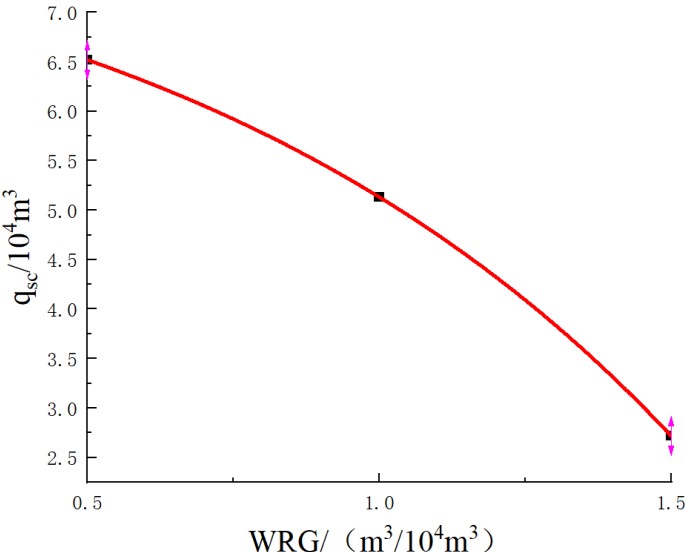

**Figure 22.** Unobstructed flow rate variation curve with changes in WRG considering the impact of water saturation on the TPG and stress sensitivity effects.

## 5. Summary

This study delves into the intricate permeation mechanisms of dense, highly water-saturated gas reservoirs through comprehensive experimentation. As the water saturation increases, the slip factor of gas gradually diminishes, becoming negligible at a water saturation of 40%. This suggests that, under conditions of elevated water saturation, the gas flow tends towards Darcy flow, no longer exhibiting characteristics of non-Darcy flow. The dense gas reservoirs exhibit pronounced stress sensitivity, with the stress sensitivity coefficient escalating with rising water saturation, particularly becoming more pronounced beyond 40% water saturation. This observation underscores that, under conditions of high water saturation, the mechanical properties of rocks are more susceptible to stress influences, necessitating a deeper exploration and analysis to comprehend and effectively manage this stress sensitivity. Upon exceeding a water saturation of 30%, dense water-saturated gas reservoirs manifest a TPG, intensifying with the elevation of water saturation. This trend becomes notably significant, particularly beyond a water saturation of 40%. This result highlights the challenges in initiating gas flow within reservoirs under high water saturation conditions, emphasizing the need to consider this factor for more accurate predictions of reservoir productivity. With the increase in water saturation, the relative permeability of gas in tight sandstone gas reservoirs sharply declines, reaching close to zero when water saturation exceeds 80%. This implies that under conditions of high water saturation, gas infiltration becomes nearly impossible, holding significant practical implications for the production and development of gas reservoirs. Addressing this challenge requires the implementation of corresponding technological measures.

Through a comprehensive analysis and summary of the experimental results, this study provided a clear definition of tight water-bearing gas reservoirs and delved into the different regional flow patterns. Given the heterogeneity and unique flow characteristics of tight sandstone gas reservoirs, various theories and methods were employed, including coordinate transformation, the elliptical coordinate theory, the perturbation ellipse theory, the pressure superposition principle, and the water–gas similarity principle, to establish a zoning productivity prediction model. This model takes into account multiple factors affecting productivity, including heterogeneity, the impact of gas well water production on productivity, and the different flow characteristics exhibited by gas wells in different regions. Using the newly developed method for predicting the production capacity of fractured wells in tight gas reservoirs, this study conducted analyses on the effects of regional differences in stress sensitivity, dynamic TPG, the number of fractures, and the water–gas ratio on well productivity.

In Region I, the high-speed non-Darcy flow within the fractures and the effects of stress sensitivity were thoroughly taken into account. This region is crucial for reservoir production since the high permeability of the fractures significantly impacts productivity. In Region II, particular attention was given to the stress sensitivity effects due to elliptical non-Darcy flow caused by fracture boundary drainage and the initiation pressure gradient effects experienced by the gas flow. The characteristic of this region lies in the significant influence of matrix stress sensitivity on productivity. Finally, in Region III, the effects of reservoir stress sensitivity and the TPG experienced by gas flow were comprehensively considered. This region requires special attention to reservoir characteristics because reservoir stress sensitivity significantly affects reservoir productivity.

This article established a mathematical model for the two-phase flow of gas and water in tight gas reservoirs. Through specific calculation examples, the practicality and accuracy of the model were verified, providing an important theoretical foundation and engineering guidance for the production capacity prediction of unconventional oil and gas resources.

**Author Contributions:** Drafting of the manuscript and formal analysis, B.W.; writing—review and editing, R.S.; methodology and data curation, Z.Z. and J.D.; supervision and project administration, X.N. and P.N.; funding acquisition, D.-t.D. and J.X. All authors have read and agreed to the published version of the manuscript.

**Funding:** This research was funded by the National Natural Science Foundation of China (52004217; 52104022), Scientific Research Program Funded by Shaanxi Provincial Education Department (Program No. 22JS030). We would like to express our appreciation to the other members of the laboratory for the help provided in experiments and language editing.

**Data Availability Statement:** Data are contained within the article.

**Conflicts of Interest:** Author Benchi Wei was studied by the Xi'an Shiyou University. Authors Zonghui Zhang and Jingchen Ding were employed by the company Sinopec North China Company. The remaining authors declare that the research was conducted in the absence of any commercial or financial relationships that could be construed as a potential conflict of interest.

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
