# Peer review of "Zoning Productivity Calculation Method of Fractured Horizontal Wells in High-Water-Cut Tight Sandstone Gas Reservoirs under Complex Seepage Conditions"

_processes, doi:10.3390/pr11123308_

Round 1
Reviewer 1 Report
Comments and Suggestions for Authors
The recent submitted manuscript titled “Zoning Productivity Calculation Method of Fractured Horizontal Wells in High-Water-Cut Tight Sandstone Gas Reservoir under Complex Seepage Conditions.” deals with the core samples from a tight sandstone gas reservoir in the 20 Ordos Basin to establish the variations in its permeability mechanism at different water saturation levels. Several experimental results presented by authors, such as the gas slip factor in tight water-bearing gas reservoirs, the stress sensitivity coefficient and the threshold pressure gradient (TPG), all these to established a zoning productivity calculation method of fractured horizontal wells in high-water-cut tight sandstone gas reservoir under complex seepage conditions and through utilizing different calculations model
I have found that the MS includes an interesting topic, a few methods have been integrated in a organized way. However, I believe the MS could be in a better scientific framework if the authors modify the MS according to reviewers feedback and comments, so kindly read my comments and modification as well and for detailed explanation see the new PDF version of the modified MS:
1. it is best to avoid personal pronouns in your academic writing even when it is personal opinion from the authors. Your findings might be underestimated by reader, …. I saw several pronouns please remove them.
2. In introduction, page 2, line 45, this statement “During the process of development and enhancement…” is incomplete.
3. In introduction, page 2, line 54-56, “At the pipe wall, gas molecules maintain certain movement states and undergo directional motion along the wall through momentum exchange, leading to directional flow of adjacent gas molecules.” cite it by source `s` at least to clarify the previous workers.
4. In introduction, page 2, line 71-83, you write the name of workers and cite them at the end, try to modify the sentences, remove the names or put years after the name of workers, for more information see the new my comments in the attached file.
5. In Materials and Methods, page 3, lines 102-103, how many samples were used? Modify the text accordingly.
6. In Materials and Methods, page 3-6, lines 110-232, the authors sometimes named the well number as core sample numbers, like X1 and 5-1/57 are represented well number and core samples, respectively, while in table 2 is completely opposite, and table 3 is so far different, this should be homogenize and clearly present because results will simultaneously influence on your arguments in discussion, abstract, conclusion parts.
7. In Materials and Methods, for measuring WS, petro-physics, slip factor….why sometimes using 4 cores and in places using 9 cores?
8. Put the units wherever is necessary.
9. In WHOLE DISCUSSION PART, the authors only based on their results without mentioned or referring to any previous scholars, which are the most important part for supporting their argument or not, and to define their novelty in what direction
10. Another strong indicator to present his case coherently, the authors have to add a new section, or heading, petro-physical or core description part, where the authors should describe their core samples, figure are very necessary to be there. Because the homogeneity degree in sandstone core sample could influence the whole measurement.
11. Finally, why the authors did not measure the porosity and porous media?
All the best

Comments on the Quality of English LanguageMinor revision are required..
Author Response
Dear Editor,
Thank you for your correspondence and the review of our manuscript. We greatly appreciate the valuable feedback provided by the reviewers, which has been immensely helpful for our research.
During the review process, we carefully examined the comments from the reviewers and addressed each suggestion with detailed responses and revisions. In addition, we have improved other aspects of the manuscript.
We believe that with these modifications, our manuscript has been significantly improved in terms of content and quality, aligning it more closely with the requirements of your esteemed journal. We hope that our efforts meet your and the reviewers' approval. Should you require additional information or have further suggestions, please feel free to contact us.
Once again, we sincerely appreciate your and the reviewers' diligent work and valuable insights.
Best regards.

Reviewer 2 Report
Comments and Suggestions for Authors
The manuscript of Wei et al., "Zoning Productivity Calculation Method of Fractured Horizontal Wells in High-Water-Cut Tight Sandstone Gas Reservoir under Complex Seepage Conditions," is an article concerning the impact of water content on the permeability mechanism of tight gas reservoirs. The authors selected core samples from a tight sandstone gas reservoir in the Ordos Basin to investigate the variations in its permeability mechanism at different water saturation levels. The study established a zoning productivity calculation method for fractured horizontal wells in high-water-cut tight sandstone gas reservoirs under complex seepage conditions. The paper itself is very interesting and informative. However, there are also additions/revisions that I feel could improve this paper. A more detailed list of my thoughts follows below.
1. Many methods and equations are shown in Section 3, which, instead, should be moved to Section 2.
2. Lots of core samples from an actual tight shale gas reservoir.were selected to conduct the mechanism analysis. Please add some information about the background of this tight shale gas reservoir for clarity.
3. The relationship in Fig.5 is only based on four measurements. How to guarantee the robustness of this relationship? Please briefly clarify it.
4. Page 14, Line 307, “This chapter” should be “This section”.
5. Page 14, Line 392-394, “This section may be divided by subheadings. It should provide a concise and precise description of the experimental results, their interpretation, as well as the experimental conclusions that can be drawn.” Please double-check this information and make some revisions.
6. Page 28, Line 655, “The viscosity of water is 1 MPa.s”, Please examine the unit.
7. The contents of this paper are a little redundant, making it like a report. Please reorganize the framework of this paper and emphasize the novelty.
Reviewer 3 Report
Comments and Suggestions for Authors
Dear Authors
after reviewing your work, the following comments are to be considered:
1. Clarify within the text especially in the abstract and discussion, the origin of the fractures are they natural or hydraulic? or both? this not clear within the entire text.
2. what is the type of the fractures? size? you mentioned only vertical? any other directions?
3. Do you have any visual evidence on the type of fractures and porosity distribution (core imaging, microscopic or CT-tomography imaging?
3. The whole paragraph between lines 464 and 478 is repeated twice word by word. Remove and rephrase.
all the best
Reviewer
Reviewer 4 Report
Comments and Suggestions for Authors
This manuscript presents an experimental study centered on the influence of the water saturation content on the permeability of tight gas reservoirs. The experimental procedure permitted to estimate the gas slip factor in core samples with different levels of water saturation.
While the manuscript provides several analyzes of interesting experimental results, the manuscript structure is difficult to follow for an independent reader. I think that the manuscript is not ready for publication in a high-quality peer reviewed journal. and still requires a lot of editing work. I think that the quality of the presentation, the scientific soundness, and the interest to the readers is Low. For those reasons, my recommendation is to reject the manuscript.
Some recommendations are listed below:
1. The Introduction section seems confusing. There are some critical terms, needed for gaining a better understanding about the research, that the readers might not fully understand. For example, it would be ideal to clarify the following terms used in the Introduction section: dual-phase hydraulic fracturing productivity (lines 48 - 49), single-phase or two-phase factors (lines 84-85), conventional productivity models (line 87).
2. Please explain how do you define the stress sensitivity coefficient (line 92).
3. I am having a hard time trying to figure out the structure of the manuscript. For example, Line 84 starts with connection words (In summary), but then there are some sentences adding new arguments.
4. Please explain the objectives of the work.
5. Please provide a brief explanation about the next sections.
6. The Materials and Method section seems interesting, but unfortunately lacks some critical explanations. For instance, could you please clarify the experimental steps?
7. Which are the control variables and which are the estimated variables?
8. I think that the slip factor is calculated from the experiments, but I am not 100% sure. Could you please explain which formula is being used to calculate the slippage factor?
9. What do you mean by conventional permeability in Table 1?
10. The content shown in pages 4, 5, and 6 seems to belong more to a discussion of Results rather than a "Materials and Methods" section. Please check if this information belongs to the Results section.
11. Notice that section 1 refers to the Introduction, section 2 is named “Materials and Method”, but section 3 is named “Derivation of Productivity Equation for Fractured Horizontal Wells in High-Water Cut Tight Gas Reservoir”. I don’t understand this structure. I believe that this type of manuscript structure makes more difficult to understand the paper.
12. There seems to be another topic emerging from Line 370 to line 670 that seems to be a new topic (model?), which is completely different in relation to the experimental work. I am having a hard time trying to understand how the section “3.2. Zoning productivity physicochemical mode of fractured horizontal wells in high-water-cut tight gas reservoir under complex seepage conditions” relates to the previous sections.
13. At this point, I don’t think that I understand much about how each part of the manuscript relates with each other. In other words, the manuscript structure is not clear to me.
14. My recommendation is to re-structure the paper. Please start by clarifying the work objectives. Please follow a more standard manuscript structure. Please provide sufficient information in each section.
15. Section 4 is a Sensitivity Analysis, but section 5 contains “Summary, Discussion, Conclusion”. This is very unusual. Please split section 5 into separated sections. By the way, I don’t understand why there is a Summary subsection next to a Conclusion subsection.
Round 2
Reviewer 1 Report
Comments and Suggestions for Authors
The present manuscript is good enough and no needs to be modified
Good luck
Reviewer 2 Report
Comments and Suggestions for Authors
I have carefully checked the modified manuscript and the response to the reviewers. I think the author has properly addressed all the comments, which in turn greatly improves the quality of the paper. Therefore, I would like to recommend this manuscript for publication.
Reviewer 4 Report
Comments and Suggestions for Authors
The authors addressed the recommendations listed in the previous round. I don’t have any further comment this time. Please make sure to correct minor details about the readability of the document.